# Risk-Averse Model Uncertainty for Distributionally Robust Safe Reinforcement Learning

**James Queeney**[*]
Division of Systems Engineering
Boston University
`jqueeney@bu.edu`

**Mouhacine Benosman**
Mitsubishi Electric Research Laboratories
`benosman@merl.com`

## Abstract

Many real-world domains require safe decision making in uncertain environments. In this work, we introduce a deep reinforcement learning framework for approaching this important problem. We consider a distribution over transition models, and apply a risk-averse perspective towards model uncertainty through the use of coherent distortion risk measures. We provide robustness guarantees for this framework by showing it is equivalent to a specific class of distributionally robust safe reinforcement learning problems. Unlike existing approaches to robustness in deep reinforcement learning, however, our formulation does not involve minimax optimization. This leads to an efficient, model-free implementation of our approach that only requires standard data collection from a single training environment. In experiments on continuous control tasks with safety constraints, we demonstrate that our framework produces robust performance and safety at deployment time across a range of perturbed test environments.

## 1 Introduction

In many real-world decision making applications, it is important to satisfy safety requirements while achieving a desired goal. In addition, real-world environments often involve uncertain or changing conditions. Therefore, in order to reliably deploy data-driven decision making methods such as deep reinforcement learning (RL) in these settings, they must deliver *robust performance and safety* even in the presence of uncertainty. Recently, techniques have been developed to handle safety constraints within the deep RL framework [3, 27, 28, 41, 44, 47, 55], but these safe RL algorithms only focus on performance and safety in the training environment. They do not consider uncertainty about the true environment at deployment time due to unknown disturbances or irreducible modeling errors, which we refer to as *model uncertainty*. In this work, we introduce a framework that incorporates model uncertainty into safe RL. In order for our framework to be useful, we emphasize the importance of (i) an efficient deep RL implementation during training and (ii) robustness guarantees on performance and safety upon deployment.

Existing robust RL methods address the issue of model uncertainty, but they can be difficult to implement and are not always suitable in real-world decision making settings. Robust RL focuses on worst-case environments in an uncertainty set, which requires solving complex minimax optimization problems throughout training. This is typically approximated in a deep RL setting through direct interventions with a learned adversary [38, 46, 49], or through the use of parametric uncertainty with multiple simulated training environments [31, 32, 39]. However, we do not always have access to fast, high-fidelity simulators for training [10, 33, 54]. In these cases, we must be able to incorporate robustness to model uncertainty without relying on multiple training environments or potentially dangerous adversarial interventions, as real-world data collection may be necessary.

---

[*]Work partly done during an internship at Mitsubishi Electric Research Laboratories.

37th Conference on Neural Information Processing Systems (NeurIPS 2023).

A more informative way to represent model uncertainty is to instead consider a distribution over potential environments. Domain randomization [37] collects training data from a range of environments by randomizing across parameter values in a simulator, and optimizes for average performance. This approach to model uncertainty avoids minimax formulations and works well in practice [5], but lacks robustness guarantees. In addition, domain randomization focuses on parametric uncertainty, which still requires detailed simulator access and domain knowledge to define the training distribution.

In this work, we introduce a general approach to safe RL in the presence of model uncertainty that addresses the main shortcomings of existing methods. In particular, we consider a distribution over potential environments, and apply a *risk-averse perspective towards model uncertainty*. Through the use of coherent distortion risk measures, this leads to a safe RL framework with robustness guarantees that does not involve difficult minimax formulations. Using this framework, we show how we can learn safe policies that are robust to model uncertainty, without the need for detailed simulator access or adversarial interventions during training. Our main contributions are as follows:

1. We reformulate the safe RL problem to incorporate a risk-averse perspective towards model uncertainty through the use of coherent distortion risk measures, and we introduce the corresponding Bellman operators.

2. From a theoretical standpoint, we provide robustness guarantees for our framework by showing it is equivalent to a specific class of distributionally robust safe RL problems.

3. We propose an efficient deep RL implementation that avoids the difficult minimax formulation present in robust RL and only uses data collected from a single training environment.

4. We demonstrate the robust performance and safety of our framework through experiments on continuous control tasks with safety constraints in the Real-World RL Suite [18, 19].

## 2 Preliminaries

**Safe reinforcement learning**   In this work, we consider RL in the presence of safety constraints. We model this sequential decision making problem as an infinite-horizon, discounted Constrained Markov Decision Process (CMDP) [4] defined by the tuple $(\mathcal{S}, \mathcal{A}, p, r, c, d_0, \gamma)$, where $\mathcal{S}$ is the set of states, $\mathcal{A}$ is the set of actions, $p : \mathcal{S} \times \mathcal{A} \to P(\mathcal{S})$ is the transition model where $P(\mathcal{S})$ represents the space of probability measures over $\mathcal{S}$, $r, c : \mathcal{S} \times \mathcal{A} \to \mathbb{R}$ are the reward function and cost function used to define the objective and constraint, respectively, $d_0 \in P(\mathcal{S})$ is the initial state distribution, and $\gamma$ is the discount rate. We focus on the setting with a single constraint, but all results can be extended to the case of multiple constraints.

We model the agent's decisions as a stationary policy $\pi : \mathcal{S} \to P(\mathcal{A})$. For a given CMDP and policy $\pi$, we write the expected total discounted rewards and costs as $J_{p,r}(\pi) = \mathbb{E}_{\tau \sim (\pi,p)} \left[ \sum_{t=0}^{\infty} \gamma^t r(s_t, a_t) \right]$ and $J_{p,c}(\pi) = \mathbb{E}_{\tau \sim (\pi,p)} \left[ \sum_{t=0}^{\infty} \gamma^t c(s_t, a_t) \right]$, respectively, where $\tau \sim (\pi, p)$ represents a trajectory sampled according to $s_0 \sim d_0$, $a_t \sim \pi(\cdot \mid s_t)$, and $s_{t+1} \sim p(\cdot \mid s_t, a_t)$. The goal of safe RL is to find a policy $\pi$ that maximizes the constrained optimization problem

$$\max_{\pi} \ J_{p,r}(\pi) \quad \text{s.t.} \quad J_{p,c}(\pi) \le B, \tag{1}$$

where $B$ is a safety budget on expected total discounted costs.

We write the corresponding state-action value functions (i.e., Q functions) for a given transition model $p$ and policy $\pi$ as $Q^{\pi}_{p,r}(s, a)$ and $Q^{\pi}_{p,c}(s, a)$, respectively. Off-policy optimization techniques [28, 55] find a policy that maximizes (1) by solving at each iteration the related optimization problem

$$\max_{\pi} \ \mathbb{E}_{s \sim \mathcal{D}} \left[ \mathbb{E}_{a \sim \pi(\cdot|s)} \left[ Q^{\pi_k}_{p,r}(s, a) \right] \right] \quad \text{s.t.} \quad \mathbb{E}_{s \sim \mathcal{D}} \left[ \mathbb{E}_{a \sim \pi(\cdot|s)} \left[ Q^{\pi_k}_{p,c}(s, a) \right] \right] \le B, \tag{2}$$

where $\pi_k$ is the current policy and $\mathcal{D}$ is a replay buffer containing data collected in the training environment. Note that $Q^{\pi}_{p,r}(s, a)$ and $Q^{\pi}_{p,c}(s, a)$ are the respective fixed points of the Bellman operators

$$\mathcal{T}^{\pi}_{p,r} Q(s, a) := r(s, a) + \gamma \mathbb{E}_{s' \sim p_{s,a}} \left[ \mathbb{E}_{a' \sim \pi(\cdot|s')} \left[ Q(s', a') \right] \right],$$

$$\mathcal{T}^{\pi}_{p,c} Q(s, a) := c(s, a) + \gamma \mathbb{E}_{s' \sim p_{s,a}} \left[ \mathbb{E}_{a' \sim \pi(\cdot|s')} \left[ Q(s', a') \right] \right].$$

**Model uncertainty in reinforcement learning** Rather than focusing on a single CMDP with transition model $p$, we incorporate uncertainty about the transition model by considering a distribution $\mu$ over models. We focus on distributions of the form $\mu = \prod_{(s,a) \in \mathcal{S} \times \mathcal{A}} \mu_{s,a}$, where $\mu_{s,a}$ represents a distribution over transition models $p_{s,a} = p(\cdot \mid s, a) \in P(\mathcal{S})$ at a given state-action pair and $\mu$ is the product over all $\mu_{s,a}$. This is known as rectangularity, and is a common assumption in the literature [11, 15, 16, 53, 56]. Note that $\mu_{s,a} \in P(\mathcal{M})$, where we write $\mathcal{M} = P(\mathcal{S})$ to denote model space. Compared to robust RL methods that apply uncertainty sets over transition models, the use of a distribution $\mu$ over transition models is a more informative way to represent model uncertainty that does not require solving for worst-case environments (i.e., *does not introduce a minimax formulation*).

In order to incorporate robustness to the choice of $\mu$, distributionally robust MDPs [53, 56] consider an ambiguity set $\mathcal{U} = \bigotimes_{(s,a) \in \mathcal{S} \times \mathcal{A}} \mathcal{U}_{s,a}$ of distributions over transition models, where $\mu_{s,a} \in \mathcal{U}_{s,a} \subseteq P(\mathcal{M})$. The goal of distributionally robust RL is to optimize the worst-case average performance across all distributions contained in $\mathcal{U}$. In this work, we will show that a risk-averse perspective towards model uncertainty defined by $\mu$ is equivalent to distributionally robust RL for appropriate choices of ambiguity sets in the objective and constraint of a CMDP. However, our use of risk measures *avoids the need to solve for worst-case distributions in $\mathcal{U}$ throughout training*.

**Risk measures** Consider the probability space $(\mathcal{M}, \mathcal{F}, \mu_{s,a})$, where $\mathcal{F}$ is a $\sigma$-algebra on $\mathcal{M}$ and $\mu_{s,a} \in P(\mathcal{M})$ defines a probability measure over $\mathcal{M}$. Let $\mathcal{Z}$ be a space of random variables defined on this probability space, and let $\mathcal{Z}^*$ be its corresponding dual space. A real-valued risk measure $\rho : \mathcal{Z} \to \mathbb{R}$ summarizes a random variable as a value on the real line. In this section, we consider cost random variables $Z \in \mathcal{Z}$ where a lower value of $\rho(Z)$ is better. We can define a corresponding risk measure $\rho^+$ for reward random variables through an appropriate change in sign, where $\rho^+(Z) = -\rho(-Z)$. Risk-sensitive methods typically focus on classes of risk measures with desirable properties [30], such as coherent risk measures [6] and distortion risk measures [17, 50].

**Definition 1** (Coherent risk measure). *A risk measure $\rho$ is a* coherent risk measure *if it satisfies monotonicity, translation invariance, positive homogeneity, and convexity.*

**Definition 2** (Distortion risk measure). *Let $g : [0, 1] \to [0, 1]$ be a non-decreasing, left-continuous function with $g(0) = 0$ and $g(1) = 1$. A distortion risk measure with respect to $g$ is defined as*

$$\rho(Z) = \int_0^1 F_Z^{-1}(u) d\tilde{g}(u),$$

*where $F_Z^{-1}$ is the inverse cumulative distribution function of $Z$ and $\tilde{g}(u) = 1 - g(1 - u)$.*

A distortion risk measure is coherent if and only if $g$ is concave [52]. In this work, we focus on the class of *coherent distortion risk measures*. We will leverage properties of coherent risk measures to provide robustness guarantees for our framework, and we will leverage properties of distortion risk measures to propose an efficient, model-free implementation that does not involve minimax optimization. See the Appendix for additional details on the properties of coherent distortion risk measures. Many commonly used risk measures belong to this class, including expectation, conditional value-at-risk (CVaR), and the Wang transform [51] for $\eta \geq 0$ which is defined by the distortion function $g_\eta(u) = \Phi(\Phi^{-1}(u) + \eta)$, where $\Phi$ is the standard Normal cumulative distribution function.

## 3 Related work

**Safe reinforcement learning** The CMDP framework is the most popular approach to safety in RL, and several deep RL algorithms have been developed to solve the constrained optimization problem in (1). These include primal-dual methods that consider the Lagrangian relaxation of (1) [41, 44, 47], algorithms that compute closed-form solutions to related or approximate versions of (1) [3, 28], and direct methods for constraint satisfaction such as the use of barriers [27] or immediate switching between the objective and constraint [55]. All of these approaches are designed to satisfy expected cost constraints for a single CMDP observed during training. In our work, on the other hand, we consider a distribution over possible transition models.

**Uncertainty in reinforcement learning** Our work focuses on irreducible uncertainty about the true environment at deployment time, which we refer to as *model uncertainty* and represent using a

distribution $\mu$ over transition models. The most popular approach that incorporates model uncertainty in this way is domain randomization [37, 48], which randomizes across parameter values in a simulator and trains a policy to maximize average performance over this training distribution. This represents a risk-neutral attitude towards model uncertainty, which has been referred to as a soft-robust approach [16]. Distributionally robust MDPs incorporate robustness to the choice of $\mu$ by instead considering a set of distributions [11, 15, 53, 56], but application of this distributionally robust framework has remained limited in deep RL as it leads to a difficult minimax formulation that requires solving for worst-case distributions over transition models.

Robust RL represents an alternative approach to model uncertainty that considers uncertainty sets of transition models [21, 34]. A major drawback of robust RL is the need to calculate worst-case environments during training, which is typically approximated through the use of parametric uncertainty with multiple training environments [31, 32, 39] or a trained adversary that directly intervenes during trajectory rollouts [38, 46, 49]. Unlike these methods, we propose a robust approach to model uncertainty based on a distribution $\mu$ over models, which does not require access to a range of simulated training environments, does not impact data collection during training, and does not involve minimax optimization problems.

In contrast to irreducible model uncertainty, *epistemic uncertainty* captures estimation error that can be reduced during training through data collection. Epistemic uncertainty has been considered in the estimation of Q functions [9, 35, 36] and learned transition models [7, 13, 22, 25, 40], and has been applied to promote both exploration and safety in a fixed MDP. Finally, risk-sensitive methods typically focus on the *aleatoric uncertainty* in RL, which refers to the range of stochastic outcomes within a single MDP. Rather than considering the standard expected value objective, they learn risk-sensitive policies over this distribution of possible outcomes in a fixed MDP [12, 24, 26, 43, 45]. Distributional RL [8] trains critics that estimate the full distribution of future returns due to aleatoric uncertainty, and risk measures can be applied to these distributional critics for risk-sensitive learning [14, 29]. We also consider the use of risk measures in our work, but different from standard risk-sensitive RL methods we apply a risk measure over *model uncertainty* instead of aleatoric uncertainty.

# 4 Risk-averse model uncertainty for safe reinforcement learning

The standard safe RL problem in (1) focuses on performance and safety in a single environment with fixed transition model $p$. In this work, however, we are interested in a distribution of possible transition models $p \sim \mu$ rather than a fixed transition model. The distribution $\mu$ provides a natural way to capture our uncertainty about the unknown transition model at deployment time. Next, we must incorporate this model uncertainty into our problem formulation. Prior methods have done this by applying the expectation operator over $\mu_{s,a}$ at every transition [16]. Instead, we adopt a risk-averse view towards model uncertainty in order to learn policies with robust performance and safety. We accomplish this by applying a coherent distortion risk measure $\rho$ *with respect to model uncertainty* at every transition.

We consider the risk-averse model uncertainty (RAMU) safe RL problem

$$\max_{\pi} \ J_{\rho^+,r}(\pi) \quad \text{s.t.} \quad J_{\rho,c}(\pi) \le B, \tag{3}$$

where we use $\rho^+$ and $\rho$ to account for reward and cost random variables, respectively, and we apply these coherent distortion risk measures over $p_{s,a} \sim \mu_{s,a}$ at every transition to define

$$J_{\rho^+,r}(\pi) := \mathop{\mathbb{E}}_{s \sim d_0} \left[ \mathop{\mathbb{E}}_{a \sim \pi(\cdot|s)} \left[ r(s,a) + \gamma \mathop{\rho^+}_{p_{s,a} \sim \mu_{s,a}} \left( \mathop{\mathbb{E}}_{s' \sim p_{s,a}} \left[ \mathop{\mathbb{E}}_{a' \sim \pi(\cdot|s')} [r(s',a') + \dots] \right] \right) \right] \right],$$

$$J_{\rho,c}(\pi) := \mathop{\mathbb{E}}_{s \sim d_0} \left[ \mathop{\mathbb{E}}_{a \sim \pi(\cdot|s)} \left[ c(s,a) + \gamma \mathop{\rho}_{p_{s,a} \sim \mu_{s,a}} \left( \mathop{\mathbb{E}}_{s' \sim p_{s,a}} \left[ \mathop{\mathbb{E}}_{a' \sim \pi(\cdot|s')} [c(s',a') + \dots] \right] \right) \right] \right].$$

The notation $\rho^+_{p_{s,a} \sim \mu_{s,a}}(\,\cdot\,)$ and $\rho_{p_{s,a} \sim \mu_{s,a}}(\,\cdot\,)$ emphasize that the stochasticity of the random variables are with respect to the transition models sampled from $\mu_{s,a}$. Note that we still apply expectations over the aleatoric uncertainty of the CMDP (i.e., the randomness associated with a stochastic transition model and stochastic policy), while being risk-averse with respect to model uncertainty. Because we are interested in learning policies that achieve robust performance *and* robust safety at deployment

time, we apply this risk-averse perspective to model uncertainty in both the objective and constraint of (3).

We write the corresponding RAMU reward and cost Q functions as $Q_{\rho^+,r}^\pi(s,a)$ and $Q_{\rho,c}^\pi(s,a)$, respectively. Similar to the standard safe RL setting, we can apply off-policy techniques to solve the RAMU safe RL problem in (3) by iteratively optimizing

$$\max_\pi \; \mathbb{E}_{s\sim\mathcal{D}}\left[\mathbb{E}_{a\sim\pi(\cdot|s)}\left[Q_{\rho^+,r}^{\pi_k}(s,a)\right]\right] \quad \text{s.t.} \quad \mathbb{E}_{s\sim\mathcal{D}}\left[\mathbb{E}_{a\sim\pi(\cdot|s)}\left[Q_{\rho,c}^{\pi_k}(s,a)\right]\right] \leq B. \tag{4}$$

Therefore, we have replaced the standard Q functions for a fixed transition model $p$ in (2) with our RAMU Q functions in (4).

We can write the RAMU Q functions recursively as

$$Q_{\rho^+,r}^\pi(s,a) = r(s,a) + \gamma \; \rho_{p_{s,a}\sim\mu_{s,a}}^+ \left( \mathbb{E}_{s'\sim p_{s,a}}\left[\mathbb{E}_{a'\sim\pi(\cdot|s')}\left[Q_{\rho^+,r}^\pi(s',a')\right]\right]\right),$$

$$Q_{\rho,c}^\pi(s,a) = c(s,a) + \gamma \; \rho_{p_{s,a}\sim\mu_{s,a}} \left( \mathbb{E}_{s'\sim p_{s,a}}\left[\mathbb{E}_{a'\sim\pi(\cdot|s')}\left[Q_{\rho,c}^\pi(s',a')\right]\right]\right).$$

These recursive definitions motivate corresponding RAMU Bellman operators.

**Definition 3** (RAMU Bellman operators). *For a given policy $\pi$, the* RAMU Bellman operators *are defined as*

$$\mathcal{T}_{\rho^+,r}^\pi Q(s,a) := r(s,a) + \gamma \; \rho_{p_{s,a}\sim\mu_{s,a}}^+ \left( \mathbb{E}_{s'\sim p_{s,a}}\left[\mathbb{E}_{a'\sim\pi(\cdot|s')}\left[Q(s',a')\right]\right]\right),$$

$$\mathcal{T}_{\rho,c}^\pi Q(s,a) := c(s,a) + \gamma \; \rho_{p_{s,a}\sim\mu_{s,a}} \left( \mathbb{E}_{s'\sim p_{s,a}}\left[\mathbb{E}_{a'\sim\pi(\cdot|s')}\left[Q(s',a')\right]\right]\right).$$

Note that the RAMU Bellman operators can also be interpreted as applying a coherent distortion risk measure over standard Bellman targets, which are random variables with respect to the transition model $p_{s,a}\sim\mu_{s,a}$ for a given state-action pair.

**Lemma 1.** *The RAMU Bellman operators can be written in terms of standard Bellman operators as*

$$\mathcal{T}_{\rho^+,r}^\pi Q(s,a) = \rho_{p_{s,a}\sim\mu_{s,a}}^+ \left(\mathcal{T}_{p,r}^\pi Q(s,a)\right), \quad \mathcal{T}_{\rho,c}^\pi Q(s,a) = \rho_{p_{s,a}\sim\mu_{s,a}} \left(\mathcal{T}_{p,c}^\pi Q(s,a)\right). \tag{5}$$

*Proof.* The results follow from the definitions of $\mathcal{T}_{p,r}^\pi$ and $\mathcal{T}_{p,c}^\pi$, along with the translation invariance and positive homogeneity of coherent distortion risk measures. See the Appendix for details. $\square$

In the next section, we show that $\mathcal{T}_{\rho^+,r}^\pi$ and $\mathcal{T}_{\rho,c}^\pi$ are contraction operators, so we can apply standard temporal difference learning techniques to learn the RAMU Q functions $Q_{\rho^+,r}^\pi(s,a)$ and $Q_{\rho,c}^\pi(s,a)$ that are needed for our RAMU policy update in (4).

## 5   Robustness guarantees

Intuitively, our risk-averse perspective places more emphasis on potential transition models that result in higher costs or lower rewards under the current policy, which should result in learning safe policies that are robust to model uncertainty. Next, we formalize the robustness guarantees of our RAMU framework by showing it is equivalent to a distributionally robust safe RL problem for appropriate choices of ambiguity sets.

**Theorem 1.** *The RAMU safe RL problem in* (3) *is equivalent to the distributionally robust safe RL problem*

$$\max_\pi \; \inf_{\beta\in\mathcal{U}^+} \mathbb{E}_{p\sim\beta}\left[J_{p,r}(\pi)\right] \quad \text{s.t.} \quad \sup_{\beta\in\mathcal{U}} \mathbb{E}_{p\sim\beta}\left[J_{p,c}(\pi)\right] \leq B \tag{6}$$

*with ambiguity sets $\mathcal{U}^+ = \bigotimes_{(s,a)\in\mathcal{S}\times\mathcal{A}}\mathcal{U}_{s,a}^+$ and $\mathcal{U} = \bigotimes_{(s,a)\in\mathcal{S}\times\mathcal{A}}\mathcal{U}_{s,a}$, where*

$$\mathcal{U}_{s,a}^+, \mathcal{U}_{s,a} \subseteq \{\beta_{s,a} \in P(\mathcal{M}) \mid \beta_{s,a} = \xi_{s,a}\mu_{s,a}, \; \xi_{s,a} \in \mathcal{Z}^*\}$$

*are sets of feasible reweightings of $\mu_{s,a}$ with $\xi_{s,a}$ that depend on the choice of $\rho^+$ and $\rho$, respectively.*

*Proof.* Using duality results for coherent risk measures [42], we show that the RAMU Bellman operators $\mathcal{T}_{\rho^+,r}^\pi$ and $\mathcal{T}_{\rho,c}^\pi$ are equivalent to distributionally robust Bellman operators [53, 56] with ambiguity sets of distributions $\mathcal{U}^+$ and $\mathcal{U}$, respectively. The RAMU Q functions are the respective fixed points of these Bellman operators, so they can be written as distributionally robust Q functions. Finally, by averaging over initial states and actions, we see that (3) is equivalent to (6). See the Appendix for details. $\square$

Theorem 1 shows that the application of $\rho^+$ and $\rho$ at every timestep are equivalent to solving distributionally robust optimization problems over the ambiguity sets of distributions $\mathcal{U}^+$ and $\mathcal{U}$, respectively. This can be interpreted as adversarially reweighting $\mu_{s,a}$ with $\xi_{s,a}$ at every state-action pair. Note that worst-case distributions appear in both the objective and constraint of (6), so any policy trained with our RAMU framework is guaranteed to deliver robust performance *and* robust safety. The level of robustness depends on the choice of $\rho^+$ and $\rho$, which determine the structure and size of the corresponding ambiguity sets based on their dual representations [42].

In addition, because (3) is equivalent to a distributionally robust safe RL problem according to Theorem 1, we can leverage existing results for distributionally robust MDPs [53, 56] to show that $\mathcal{T}_{\rho^+,r}^\pi$ and $\mathcal{T}_{\rho,c}^\pi$ are contraction operators.

**Corollary 1.** *The RAMU Bellman operators $\mathcal{T}_{\rho^+,r}^\pi$ and $\mathcal{T}_{\rho,c}^\pi$ are $\gamma$-contractions in the sup-norm.*

*Proof.* Apply results from Xu and Mannor [53] and Yu and Xu [56]. See the Appendix for details. $\square$

Therefore, we have that $Q_{\rho^+,r}^\pi(s,a)$ and $Q_{\rho,c}^\pi(s,a)$ can be interpreted as distributionally robust Q functions by Theorem 1, and we can apply standard temporal difference methods to learn these RAMU Q functions as a result of Corollary 1. Importantly, Theorem 1 demonstrates the robustness properties of our RAMU framework, *but it is not used to implement our approach*. Directly implementing (6) would require solving for adversarial distributions over transition models throughout training. Instead, our framework provides the same robustness, but the use of risk measures leads to an efficient deep RL implementation as we describe in the following section.

## 6 Model-free implementation with a single training environment

The RAMU policy update in (4) takes the same form as the standard safe RL update in (2), except for the use of $Q_{\rho^+,r}^\pi(s,a)$ and $Q_{\rho,c}^\pi(s,a)$. Because our RAMU Bellman operators are contractions, we can learn these RAMU Q functions by applying standard temporal difference loss functions that are used throughout deep RL. In particular, we consider parameterized critics $Q_{\theta_r}$ and $Q_{\theta_c}$, and we optimize their parameters during training to minimize the loss functions

$$\mathcal{L}^+(\theta_r) = \mathop{\mathbb{E}}_{(s,a)\sim\mathcal{D}}\left[\left(Q_{\theta_r}(s,a) - \hat{\mathcal{T}}_{\rho^+,r}^\pi \bar{Q}_{\theta_r}(s,a)\right)^2\right],$$

$$\mathcal{L}(\theta_c) = \mathop{\mathbb{E}}_{(s,a)\sim\mathcal{D}}\left[\left(Q_{\theta_c}(s,a) - \hat{\mathcal{T}}_{\rho,c}^\pi \bar{Q}_{\theta_c}(s,a)\right)^2\right],$$

where $\hat{\mathcal{T}}_{\rho^+,r}^\pi$ and $\hat{\mathcal{T}}_{\rho,c}^\pi$ represent sample-based estimates of the RAMU Bellman operators applied to target Q functions denoted by $\bar{Q}$. Therefore, we must be able to efficiently estimate the RAMU Bellman targets, which involve calculating coherent distortion risk measures that depend on the distribution $\mu_{s,a}$.

**Sample-based estimation of risk measures** Using the formulation of our RAMU Bellman operators from Lemma 1, we can leverage properties of distortion risk measures to efficiently estimate the results in (5) using sample-based weighted averages of standard Bellman targets. For $n$ transition models $p_{s,a}^{(i)}$, $i = 1,\ldots,n$, sampled independently from $\mu_{s,a}$ and sorted according to their corresponding Bellman targets, consider the weights

$$w_\rho^{(i)} = g\left(\frac{i}{n}\right) - g\left(\frac{i-1}{n}\right),$$

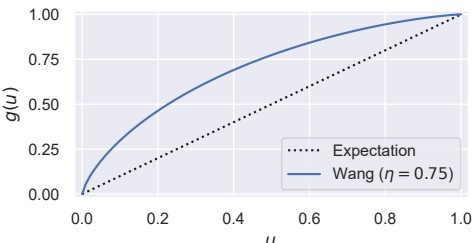 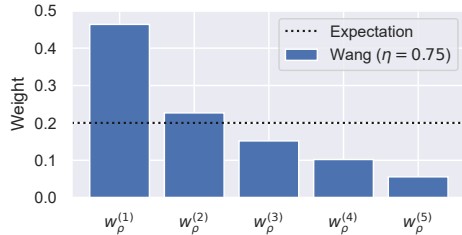

Figure 1: Coherent distortion risk measures used in RAMU experiments. Left: Distortion function $g$. Right: Weights for sample-based estimates in (7) when $n = 5$.

where $g$ defines the distortion risk measure $\rho$ according to Definition 2. See Figure 1 for the distortion functions and weights associated with the risk measures used in our experiments. Then, from Jones and Zitikis [23] we have that

$$\sum_{i=1}^{n} w_{\rho^+}^{(i)} \mathcal{T}_{p^{(i)},r}^{\pi} Q(s,a), \quad \sum_{i=1}^{n} w_{\rho}^{(i)} \mathcal{T}_{p^{(i)},c}^{\pi} Q(s,a),$$

are consistent estimators of the results in (5), where $\mathcal{T}_{p^{(i)},r}^{\pi} Q(s,a)$ are sorted in ascending order and $\mathcal{T}_{p^{(i)},c}^{\pi} Q(s,a)$ are sorted in descending order. Finally, we can replace $\mathcal{T}_{p^{(i)},r}^{\pi} Q(s,a)$ and $\mathcal{T}_{p^{(i)},c}^{\pi} Q(s,a)$ with the standard unbiased sample-based estimates

$$\hat{\mathcal{T}}_{p^{(i)},r}^{\pi} Q(s,a) = r(s,a) + \gamma Q(s',a'), \quad \hat{\mathcal{T}}_{p^{(i)},c}^{\pi} Q(s,a) = c(s,a) + \gamma Q(s',a'),$$

where $s' \sim p_{s,a}^{(i)}$ and $a' \sim \pi(\cdot \mid s')$. This leads to the sample-based estimates

$$\hat{\mathcal{T}}_{\rho^+,r}^{\pi} Q(s,a) = \sum_{i=1}^{n} w_{\rho^+}^{(i)} \hat{\mathcal{T}}_{p^{(i)},r}^{\pi} Q(s,a), \quad \hat{\mathcal{T}}_{\rho,c}^{\pi} Q(s,a) = \sum_{i=1}^{n} w_{\rho}^{(i)} \hat{\mathcal{T}}_{p^{(i)},c}^{\pi} Q(s,a), \quad (7)$$

which we use to train our RAMU Q functions. Note that the estimates in (7) can be computed very efficiently, which is a major benefit of our RAMU framework compared to robust RL methods. Next, we describe how we can sample models $p_{s,a}^{(i)}$, $i = 1, \ldots, n$, from $\mu_{s,a}$, and generate state transitions from these models to use in the calculation of our sample-based Bellman targets in (7).

**Generative distribution of transition models**   Note that our RAMU framework can be applied using any choice of distribution $\mu$, provided we can sample transition models $p_{s,a}^{(i)} \sim \mu_{s,a}$ and corresponding next states $s' \sim p_{s,a}^{(i)}$. In this work, we define the distribution $\mu$ over perturbed versions of a single training environment $p^{\text{train}}$, and we propose a generative approach to sampling transition models and corresponding next states that only requires data collected from $p^{\text{train}}$. By doing so, our RAMU framework achieves robust performance and safety with minimal assumptions on the training process, and can even be applied to settings that require real-world data collection for training.

We consider a latent variable $x \sim X$, and we define a transition model $p_{s,a}(x)$ for every $x \sim X$ that shifts the probability of $s'$ under $p_{s,a}^{\text{train}}$ according to a perturbation function $f_x : \mathcal{S} \times \mathcal{S} \to \mathcal{S}$. This perturbation function takes as input a state transition $(s, s')$, and outputs a perturbed next state $\tilde{s}'$ that depends on the latent variable $x \sim X$. Therefore, a distribution over latent space implicitly defines a distribution $\mu_{s,a}$ over perturbed versions of $p_{s,a}^{\text{train}}$. In order to obtain the next state samples needed to compute Bellman target estimates in (7), we sample latent variables $x \sim X$ and apply $f_x$ to the state transition observed in the training environment. We have that $s' \sim p_{s,a}^{\text{train}}$ for data collected in the training environment, so $\tilde{s}' = f_x(s, s')$ represents the corresponding sample from the perturbed transition model $p_{s,a}(x)$.

In our experiments, we consider a simple implementation for the common case where $\mathcal{S} = \mathbb{R}^d$. We use uniformly distributed latent variables $x \sim U([-2\epsilon, 2\epsilon]^d)$, and we define the perturbation function as

$$f_x(s, s') = s + (s' - s)(1 + x),$$

---

**Algorithm 1** Risk-Averse Model Uncertainty for Safe RL

---

**Input:** policy $\pi_0$, critics $Q_{\theta_r}, Q_{\theta_c}$, risk measures $\rho^+, \rho$, latent random variable $X$

**for** $k = 0, 1, 2, \ldots$ **do**

    Collect data $\tau \sim (\pi_k, p^{\text{train}})$ and store it in $\mathcal{D}$

    **for** $K$ updates **do**

        Sample batch of data $(s, a, r, c, s') \sim \mathcal{D}$

        Sample $n$ latent variables $x_i \sim X$ per data point, and compute next state samples $f_{x_i}(s, s')$

        Calculate Bellman targets in (7), and update critics $Q_{\theta_r}, Q_{\theta_c}$ to minimize $\mathcal{L}^+(\theta_r), \mathcal{L}(\theta_c)$

        Update policy $\pi$ according to (4)

    **end for**

**end for**

---

where all operations are performed per-coordinate. Therefore, the latent variable $x \sim U([-2\epsilon, 2\epsilon]^d)$ can be interpreted as the percentage change in each dimension of a state transition observed in the training environment, where the average magnitude of the percentage change is $\epsilon$. The hyperparameter $\epsilon$ determines the distribution $\mu_{s,a}$ over transition models, where a larger value of $\epsilon$ leads to transition models that vary more significantly from the training environment. The structure of $f_x$ provides an intuitive, scale-invariant meaning for the hyperparameter $\epsilon$, which makes it easy to tune in practice. This choice of distribution $\mu_{s,a}$ captures general uncertainty in the training environment, without requiring specific domain knowledge of potential disturbances.

**Algorithm** We summarize the implementation of our RAMU framework in Algorithm 1. Given data collected in a single training environment, we can efficiently calculate the sample-based RAMU Bellman targets in (7) by (i) sampling from a latent variable $x \sim X$, (ii) computing the corresponding next state samples $f_x(s, s')$, and (iii) sorting the standard Bellman estimates that correspond to these sampled transition models. Given the sample-based RAMU Bellman targets, updates of the critics and policy have the same form as in standard deep safe RL algorithms. Therefore, *our RAMU framework can be easily combined with many popular safe RL algorithms to incorporate model uncertainty with robustness guarantees*, using only a minor change to the estimation of Bellman targets that is efficient to implement in practice.

## 7 Experiments

In order to evaluate the performance and safety of our RAMU framework, we conduct experiments on 5 continuous control tasks with safety constraints from the Real-World RL Suite [18, 19]: Cartpole Swingup, Walker Walk, Walker Run, Quadruped Walk, and Quadruped Run. Each task has a horizon length of 1,000 with $r(s, a) \in [0, 1]$ and $c(s, a) \in \{0, 1\}$, and we consider a safety budget of $B = 100$. Unless noted otherwise, we train these tasks on a single training environment for 1 million steps across 5 random seeds, and we evaluate performance of the learned policies across a range of perturbed test environments via 10 trajectory rollouts. See the Appendix for information on the safety constraints and environment perturbations that we consider.

Our RAMU framework can be combined with several choices of safe RL algorithms. We consider the safe RL algorithm Constraint-Rectified Policy Optimization (CRPO) [55], and we use Maximum a Posteriori Policy Optimization (MPO) [1] as the unconstrained policy optimization algorithm in CRPO. For a fair comparison, we apply this choice of safe RL policy update in every method we consider in our experiments. We use a multivariate Gaussian policy with learned mean and diagonal covariance at each state, along with separate reward and cost critics. We parameterize our policy and critics using neural networks. See the Appendix for implementation details.[1]

We summarize the performance and safety of our RAMU framework in Table 1 and Figure 2, compared to several baseline algorithms that we discuss next. We include detailed experimental results across all perturbed test environments in the Appendix. We apply our RAMU framework using the Wang transform with $\eta = 0.75$ as the risk measure in both the objective and constraint. In order

---

[1]Code is publicly available at `https://github.com/jqueeney/robust-safe-rl`.

Table 1: Aggregate performance summary

| Algorithm | % Safe[†] | Normalized Ave.[‡] | | Rollouts Require[*] | |
| | | Reward | Cost | Adversary | Simulator |
|---|---|---|---|---|---|
| Safe RL | 51% | 1.00 | 1.00 | No | No |
| **RAMU (Wang 0.75)** | **80%** | **1.08** | **0.51** | **No** | **No** |
| RAMU (Expectation) | 74% | 1.05 | 0.67 | No | No |
| Domain Randomization | 76% | 1.14 | 0.72 | No | Yes |
| Domain Randomization (OOD) | 55% | 1.02 | 1.02 | No | Yes |
| Adversarial RL | 82% | 1.05 | 0.48 | Yes | No |

[†] Percentage of policies that satisfy the safety constraint across all tasks and test environments.

[‡] Normalized relative to the average performance of standard safe RL for each task and test environment.

[*] Denotes need for adversary or simulator during data collection (i.e., trajectory rollouts) for training.

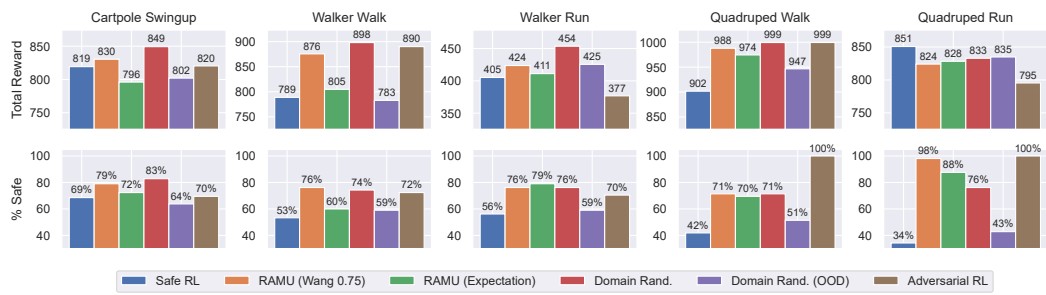

Figure 2: Performance summary by task, aggregated across perturbed test environments. Performance of adversarial RL is evaluated without adversarial interventions. Top: Total rewards averaged across test environments. Bottom: Percentage of policies across test environments that satisfy the safety constraint.

to understand the impact of being risk-averse to model uncertainty, we also consider the risk-neutral special case of our framework where expectations are applied to the objective and constraint. For our RAMU results in Table 1 and Figure 2, we specify the risk measure in parentheses. Finally, we consider $n = 5$ samples of transition models with latent variable hyperparameter $\epsilon = 0.10$ in order to calculate Bellman targets in our RAMU framework.

**Comparison to safe reinforcement learning** First, we analyze the impact of our RAMU framework compared to standard safe RL. In both cases, we train policies using data collected from a single training environment, so the only difference comes from our use of risk-averse model uncertainty to learn RAMU Q functions. By evaluating the learned policies in perturbed test environments different from the training environment, we see that our RAMU framework provides robustness in terms of both total rewards and safety. In particular, the risk-averse implementation of our algorithm leads to safety constraint satisfaction in 80% of test environments, compared to only 51% with standard safe RL. In addition, this implementation results in higher total rewards (1.08x) and lower total costs (0.51x), on average. We see in Table 1 that the use of expectations over model uncertainty (i.e., a risk-neutral approach) also improves robustness in both the objective and constraint, on average, compared to standard safe RL. However, we further improve upon the benefits observed in the risk-neutral case by instead applying a risk-averse perspective.

**Comparison to domain randomization** Next, we compare our RAMU framework to domain randomization, a popular approach that also represents model uncertainty using a distribution $\mu$ over models. Note that domain randomization considers parametric uncertainty and has the benefit of training on a range of simulated environments, while our method only collects data from a single training environment. In order to evaluate the importance of domain knowledge for defining the training distribution in domain randomization, we consider two different cases: an in-distribution

version that trains on a subset of the perturbed test environments, and an out-of-distribution (OOD) version that randomizes over a different perturbation parameter than the one varied at test time.

The results in Table 1 and Figure 2 show the importance of domain knowledge: in-distribution domain randomization leads to improved robustness compared to standard safe RL and the highest normalized average rewards (1.14x), while the out-of-distribution version provides little benefit. In both cases, however, domain randomization achieves lower levels of safety, on average, than our risk-averse formulation. In fact, we see in Figure 2 that the safety constraint satisfaction of our risk-averse formulation is at least as strong as both versions of domain randomization in 4 out of 5 tasks, *despite only training on a single environment with no specific knowledge about the disturbances at test time*. This demonstrates the key benefit of our risk-averse approach to model uncertainty.

**Comparison to adversarial reinforcement learning**   Finally, we compare our approach to adversarial RL using the action-robust PR-MDP framework [46], which randomly applies worst-case actions a percentage of the time during data collection. Although adversarial RL only collects data from a single training environment, it requires potentially dangerous adversarial interventions during training in order to provide robustness at test time. In order to apply this method to the safe RL setting, we train an adversary to maximize costs and consider a 5% probability of intervention during training. The performance of adversarial RL is typically evaluated without adversarial interventions, which requires a clear distinction between training and testing.

We see in Figure 2 that adversarial RL learns policies that achieve robust safety constraint satisfaction at test time in the Quadruped tasks. Our risk-averse formulation, on the other hand, achieves higher levels of safety in the remaining 3 out of 5 tasks, and similar levels of safety on average. Unlike adversarial RL, our RAMU framework achieves robust safety in a way that (i) does not alter the data collection process, (ii) does not require training an adversary in a minimax formulation, and (iii) does not require different implementations during training and testing. In addition, our use of a distribution over models represents a less conservative approach than adversarial RL, resulting in higher normalized average rewards as shown in Table 1.

# 8   Conclusion

We have presented a framework for safe RL in the presence of model uncertainty, an important setting for many real-world decision making applications. Compared to existing approaches to model uncertainty in deep RL, our formulation applies a risk-averse perspective through the use of coherent distortion risk measures. We show that this results in robustness guarantees, while still leading to an efficient deep RL implementation that does not involve minimax optimization problems. Importantly, our method only requires data collected from a single training environment, so it can be applied to real-world domains where high-fidelity simulators are not readily available or are computationally expensive. Therefore, our framework represents an attractive approach to safe decision making under model uncertainty that can be deployed across a range of applications.

Prior to potential deployment, it is important to understand the limitations of our proposed methodology. The robustness and safety of our RAMU framework depend on the user-defined choices of model distribution $\mu$ and risk measure $\rho$. The distribution $\mu$ defines the uncertainty over transition models, and the risk measure $\rho$ defines the level of robustness to this choice of $\mu$. In addition, our approach only considers robustness with respect to model uncertainty and safety as defined by expected total cost constraints. It would be interesting to extend our techniques to address other forms of uncertainty and other definitions of safety, including epistemic uncertainty in model-based RL, observational uncertainty, and safety-critical formulations based on sets of unsafe states.

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

# A   Properties of coherent distortion risk measures

Majumdar and Pavone [30] proposed a set of six axioms to characterize desirable properties of risk measures in the context of robotics.

A1. Monotonicity: If $Z, Z' \in \mathcal{Z}$ and $Z \leq Z'$ almost everywhere, then $\rho(Z) \leq \rho(Z')$.

A2. Translation invariance: If $\alpha \in \mathbb{R}$ and $Z \in \mathcal{Z}$, then $\rho(Z + \alpha) = \rho(Z) + \alpha$.

A3. Positive homogeneity: If $\tau \geq 0$ and $Z \in \mathcal{Z}$, then $\rho(\tau Z) = \tau \rho(Z)$.

A4. Convexity: If $\lambda \in [0, 1]$ and $Z, Z' \in \mathcal{Z}$, then $\rho(\lambda Z + (1 - \lambda)Z') \leq \lambda\rho(Z) + (1 - \lambda)\rho(Z')$.

A5. Comonotonic additivity: If $Z, Z' \in \mathcal{Z}$ are comonotonic, then $\rho(Z + Z') = \rho(Z) + \rho(Z')$.

A6. Law invariance: If $Z, Z' \in \mathcal{Z}$ are identically distributed, then $\rho(Z) = \rho(Z')$.

See Majumdar and Pavone [30] for a discussion on the intuition behind these axioms. Note that coherent risk measures [6] satisfy Axioms A1–A4, distortion risk measures [17, 50] satisfy Axioms A1–A3 and Axioms A5–A6, and coherent distortion risk measures satisfy all six axioms.

The properties of coherent risk measures also lead to a useful dual representation.

**Lemma 2** (Shapiro et al. [42]). *Let $\rho$ be a proper, real-valued coherent risk measure. Then, for any $Z \in \mathcal{Z}$ we have that*

$$\rho(Z) = \sup_{\beta_{s,a} \in \mathcal{U}_{s,a}} \mathbb{E}_{\beta_{s,a}} [Z],$$

*where $\mathbb{E}_{\beta_{s,a}} [\cdot]$ represents expectation with respect to the probability measure $\beta_{s,a} \in P(\mathcal{M})$, and*

$$\mathcal{U}_{s,a} \subseteq \{\beta_{s,a} \in P(\mathcal{M}) \mid \beta_{s,a} = \xi_{s,a}\mu_{s,a}, \ \xi_{s,a} \in \mathcal{Z}^*\}$$

*is a convex, bounded, and weakly\* closed set that depends on $\rho$.*

See Shapiro et al. [42] for a general treatment of this result.

# B   Proofs

In this section, we prove all results related to the RAMU cost Bellman operator $\mathcal{T}_{\rho,c}^\pi$. Using the fact that $\rho^+(Z) = -\rho(-Z)$ for a coherent distortion risk measure $\rho$ on a cost random variable, all results related to the RAMU reward Bellman operator follow by an appropriate change in sign.

## B.1   Proof of Lemma 1

*Proof.* Starting from the definition of $\mathcal{T}_{\rho,c}^\pi$ in Definition 3, we have that

$$\mathcal{T}_{\rho,c}^\pi Q(s,a) = c(s,a) + \gamma \rho_{p_{s,a} \sim \mu_{s,a}} \left( \mathbb{E}_{s' \sim p_{s,a}} \left[ \mathbb{E}_{a' \sim \pi(\cdot|s')} [Q(s',a')] \right] \right)$$

$$= c(s,a) + \rho_{p_{s,a} \sim \mu_{s,a}} \left( \gamma \mathbb{E}_{s' \sim p_{s,a}} \left[ \mathbb{E}_{a' \sim \pi(\cdot|s')} [Q(s',a')] \right] \right)$$

$$= \rho_{p_{s,a} \sim \mu_{s,a}} \left( c(s,a) + \gamma \mathbb{E}_{s' \sim p_{s,a}} \left[ \mathbb{E}_{a' \sim \pi(\cdot|s')} [Q(s',a')] \right] \right)$$

$$= \rho_{p_{s,a} \sim \mu_{s,a}} \left( \mathcal{T}_{p,c}^\pi Q(s,a) \right),$$

which proves the result. Note that the second equality follows from the positive homogeneity of $\rho$ (Axiom A3), the third equality follows from the translation invariance of $\rho$ (Axiom A2), and the fourth equality follows from the definition of the standard cost Bellman operator $\mathcal{T}_{p,c}^\pi$. □

## B.2   Proof of Theorem 1

*Proof.* First, we show that $\mathcal{T}_{\rho,c}^\pi$ is equivalent to a distributionally robust Bellman operator. For a given state-action pair, we apply Lemma 2 to the risk measure that appears in the formulation of $\mathcal{T}_{\rho,c}^\pi$

given by Lemma 1. By doing so, we have that

$$\mathcal{T}^\pi_{\rho,c} Q(s,a) = \underset{p_{s,a} \sim \mu_{s,a}}{\rho} \left( \mathcal{T}^\pi_{p,c} Q(s,a) \right)$$

$$= \sup_{\beta_{s,a} \in \mathcal{U}_{s,a}} \underset{p_{s,a} \sim \beta_{s,a}}{\mathbb{E}} \left[ \mathcal{T}^\pi_{p,c} Q(s,a) \right]$$

$$= c(s,a) + \gamma \sup_{\beta_{s,a} \in \mathcal{U}_{s,a}} \underset{p_{s,a} \sim \beta_{s,a}}{\mathbb{E}} \left[ \underset{s' \sim p_{s,a}}{\mathbb{E}} \left[ \underset{a' \sim \pi(\cdot|s')}{\mathbb{E}} [Q(s',a')] \right] \right],$$

where $\mathcal{U}_{s,a}$ is defined in Lemma 2. Therefore, $\mathcal{T}^\pi_{\rho,c}$ has the same form as a distributionally robust Bellman operator [53, 56] with the ambiguity set $\mathcal{U} = \bigotimes_{(s,a) \in \mathcal{S} \times \mathcal{A}} \mathcal{U}_{s,a}$. The RAMU cost Q function $Q^\pi_{\rho,c}(s,a)$ is the fixed point of $\mathcal{T}^\pi_{\rho,c}$, so it is equivalent to a distributionally robust Q function with ambiguity set $\mathcal{U}$. Using the rectangularity of $\mathcal{U}$, we can write this succinctly as

$$Q^\pi_{\rho,c}(s,a) = \sup_{\beta \in \mathcal{U}} \underset{p \sim \beta}{\mathbb{E}} \left[ Q^\pi_{p,c}(s,a) \right].$$

Then, using the definition of $J_{\rho,c}(\pi)$ we have that

$$J_{\rho,c}(\pi) = \underset{s \sim d_0}{\mathbb{E}} \left[ \underset{a \sim \pi(\cdot|s)}{\mathbb{E}} \left[ Q^\pi_{\rho,c}(s,a) \right] \right]$$

$$= \underset{s \sim d_0}{\mathbb{E}} \left[ \underset{a \sim \pi(\cdot|s)}{\mathbb{E}} \left[ \sup_{\beta \in \mathcal{U}} \underset{p \sim \beta}{\mathbb{E}} \left[ Q^\pi_{p,c}(s,a) \right] \right] \right]$$

$$= \sup_{\beta \in \mathcal{U}} \underset{p \sim \beta}{\mathbb{E}} \left[ \underset{s \sim d_0}{\mathbb{E}} \left[ \underset{a \sim \pi(\cdot|s)}{\mathbb{E}} \left[ Q^\pi_{p,c}(s,a) \right] \right] \right]$$

$$= \sup_{\beta \in \mathcal{U}} \underset{p \sim \beta}{\mathbb{E}} \left[ J_{p,c}(\pi) \right],$$

where we can move the optimization over $\mathcal{U}$ outside of the expectation operators due to rectangularity.

We can use similar techniques to show that $\mathcal{T}^\pi_{\rho^+,r}$ has the same form as a distributionally robust Bellman operator with the ambiguity set $\mathcal{U}^+ = \bigotimes_{(s,a) \in \mathcal{S} \times \mathcal{A}} \mathcal{U}^+_{s,a}$, and

$$J_{\rho^+,r}(\pi) = \inf_{\beta \in \mathcal{U}^+} \underset{p \sim \beta}{\mathbb{E}} \left[ J_{p,r}(\pi) \right].$$

Therefore, we have that the RAMU safe RL problem in (3) is equivalent to (6). $\square$

## B.3 Proof of Corollary 1

Given the equivalence of $\mathcal{T}^\pi_{\rho^+,r}$ and $\mathcal{T}^\pi_{\rho,c}$ to distributionally robust Bellman operators as shown in Theorem 1, Corollary 1 follows from results in Xu and Mannor [53] and Yu and Xu [56]. We include a proof for completeness.

*Proof.* Due to the linearity of the expectation operator, for a given $\beta_{s,a} \in \mathcal{U}_{s,a}$ we have that

$$\underset{p_{s,a} \sim \beta_{s,a}}{\mathbb{E}} \left[ \underset{s' \sim p_{s,a}}{\mathbb{E}} \left[ \underset{a' \sim \pi(\cdot|s')}{\mathbb{E}} [Q(s',a')] \right] \right] = \underset{s' \sim \bar{p}^\beta_{s,a}}{\mathbb{E}} \left[ \underset{a' \sim \pi(\cdot|s')}{\mathbb{E}} [Q(s',a')] \right],$$

where $\bar{p}^\beta_{s,a} = \mathbb{E}_{p_{s,a} \sim \beta_{s,a}}[p_{s,a}] \in P(\mathcal{S})$ represents a mixture transition model determined by $\beta_{s,a}$. Therefore, starting from the result in Theorem 1, we can write

$$\mathcal{T}^\pi_{\rho,c} Q(s,a) = c(s,a) + \gamma \sup_{\beta_{s,a} \in \mathcal{U}_{s,a}} \underset{p_{s,a} \sim \beta_{s,a}}{\mathbb{E}} \left[ \underset{s' \sim p_{s,a}}{\mathbb{E}} \left[ \underset{a' \sim \pi(\cdot|s')}{\mathbb{E}} [Q(s',a')] \right] \right]$$

$$= c(s,a) + \gamma \sup_{\bar{p}^\beta_{s,a} \in \mathcal{P}_{s,a}} \underset{s' \sim \bar{p}^\beta_{s,a}}{\mathbb{E}} \left[ \underset{a' \sim \pi(\cdot|s')}{\mathbb{E}} [Q(s',a')] \right],$$

where

$$\mathcal{P}_{s,a} = \left\{ \bar{p}^\beta_{s,a} \in P(\mathcal{S}) \mid \bar{p}^\beta_{s,a} = \underset{p_{s,a} \sim \beta_{s,a}}{\mathbb{E}} [p_{s,a}], \ \beta_{s,a} \in \mathcal{U}_{s,a} \right\}.$$

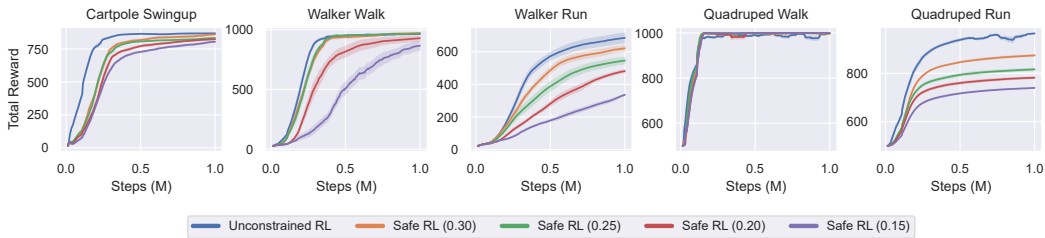

Figure 3: Hyperparameter sweep of safety coefficient. Value in parentheses represents safety coefficient used for training in safe RL. Shading denotes half of one standard error across policies.

As a result, $\mathcal{T}_{\rho,c}^{\pi}$ has the same form as a robust Bellman operator [21, 34] with the uncertainty set $\mathcal{P} = \bigotimes_{(s,a)\in\mathcal{S}\times\mathcal{A}} \mathcal{P}_{s,a}$.

Consider Q functions $Q^{(1)}$ and $Q^{(2)}$, and denote the sup-norm by

$$\|Q^{(1)} - Q^{(2)}\|_{\infty} = \sup_{(s,a)\in\mathcal{S}\times\mathcal{A}} \left| Q^{(1)}(s,a) - Q^{(2)}(s,a) \right|.$$

Fix $\epsilon > 0$ and consider $(s,a) \in \mathcal{S} \times \mathcal{A}$. Then, there exists $\bar{p}_{s,a}^{(1)} \in \mathcal{P}_{s,a}$ such that

$$\mathbb{E}_{s'\sim\bar{p}_{s,a}^{(1)}}\left[ \mathbb{E}_{a'\sim\pi(\cdot|s')}\left[ Q^{(1)}(s',a') \right] \right] \geq \sup_{\bar{p}_{s,a}^{\beta}\in\mathcal{P}_{s,a}} \mathbb{E}_{s'\sim\bar{p}_{s,a}^{\beta}}\left[ \mathbb{E}_{a'\sim\pi(\cdot|s')}\left[ Q^{(1)}(s',a') \right] \right] - \epsilon.$$

We have that

$$\mathcal{T}_{\rho,c}^{\pi}Q^{(1)}(s,a) - \mathcal{T}_{\rho,c}^{\pi}Q^{(2)}(s,a)$$

$$= \gamma\left( \sup_{\bar{p}_{s,a}^{\beta}\in\mathcal{P}_{s,a}} \mathbb{E}_{s'\sim\bar{p}_{s,a}^{\beta}}\left[ \mathbb{E}_{a'\sim\pi(\cdot|s')}\left[ Q^{(1)}(s',a') \right] \right] - \sup_{\bar{p}_{s,a}^{\beta}\in\mathcal{P}_{s,a}} \mathbb{E}_{s'\sim\bar{p}_{s,a}^{\beta}}\left[ \mathbb{E}_{a'\sim\pi(\cdot|s')}\left[ Q^{(2)}(s',a') \right] \right] \right)$$

$$\leq \gamma\left( \mathbb{E}_{s'\sim\bar{p}_{s,a}^{(1)}}\left[ \mathbb{E}_{a'\sim\pi(\cdot|s')}\left[ Q^{(1)}(s',a') \right] \right] + \epsilon - \mathbb{E}_{s'\sim\bar{p}_{s,a}^{(1)}}\left[ \mathbb{E}_{a'\sim\pi(\cdot|s')}\left[ Q^{(2)}(s',a') \right] \right] \right)$$

$$= \gamma \mathbb{E}_{s'\sim\bar{p}_{s,a}^{(1)}}\left[ \mathbb{E}_{a'\sim\pi(\cdot|s')}\left[ Q^{(1)}(s',a') - Q^{(2)}(s',a') \right] \right] + \gamma\epsilon$$

$$\leq \gamma\|Q^{(1)} - Q^{(2)}\|_{\infty} + \gamma\epsilon.$$

A similar argument can be used to show that

$$-\gamma\|Q^{(1)} - Q^{(2)}\|_{\infty} - \gamma\epsilon \leq \mathcal{T}_{\rho,c}^{\pi}Q^{(1)}(s,a) - \mathcal{T}_{\rho,c}^{\pi}Q^{(2)}(s,a),$$

so we have that

$$\left| \mathcal{T}_{\rho,c}^{\pi}Q^{(1)}(s,a) - \mathcal{T}_{\rho,c}^{\pi}Q^{(2)}(s,a) \right| \leq \gamma\|Q^{(1)} - Q^{(2)}\|_{\infty} + \gamma\epsilon.$$

By applying a supremum over state-action pairs on the left-hand side, we obtain

$$\|\mathcal{T}_{\rho,c}^{\pi}Q^{(1)} - \mathcal{T}_{\rho,c}^{\pi}Q^{(2)}\|_{\infty} \leq \gamma\|Q^{(1)} - Q^{(2)}\|_{\infty} + \gamma\epsilon.$$

Finally, since $\epsilon > 0$ was arbitrary, we have shown that $\mathcal{T}_{\rho,c}^{\pi}$ is a $\gamma$-contraction in the sup-norm. $\qquad\square$

## C  Implementation details and additional experimental results

**Safety constraints and environment perturbations**  In all of our experiments, we consider the problem of optimizing a task objective while satisfying a safety constraint. We focus on a single safety constraint corresponding to a cost function defined in the Real-World RL Suite for each task,

Table 2: Safety constraints for all tasks

| Task | Safety Constraint | Safety Coefficient |
|---|---|---|
| Cartpole Swingup | Slider Position | 0.30 |
| Walker Walk | Joint Velocity | 0.25 |
| Walker Run | Joint Velocity | 0.30 |
| Quadruped Walk | Joint Angle | 0.15 |
| Quadruped Run | Joint Angle | 0.30 |

Table 3: Perturbation ranges for test environments across domains

| Domain | Perturbation Parameter | Nominal Value | Test Range |
|---|---|---|---|
| Cartpole | Pole Length | 1.00 | $[0.75, 1.25]$ |
| Walker | Torso Length | 0.30 | $[0.10, 0.50]$ |
| Quadruped | Torso Density | 1,000 | $[500, 1,500]$ |

and we consider a safety budget of $B = 100$. The safety constraints used for each task are described in Table 2. In the Cartpole domain, costs are applied when the slider is outside of a specified range. In the Walker domain, costs are applied for large joint velocities. In the Quadruped domain, costs are applied for large joint angles. See Dulac-Arnold et al. [19] for detailed definitions of each safety constraint.

The definitions of these cost functions depend on a safety coefficient in $[0, 1]$, which determines the range of outcomes that lead to constraint violations and therefore controls how difficult it will be to satisfy safety constraints corresponding to these cost functions. As the safety coefficient decreases, the range of safe outcomes also decreases and the safety constraint becomes more difficult to satisfy. In order to consider safe RL tasks with difficult safety constraints where strong performance is still possible, we selected the value of this safety constraint in the range of $[0.15, 0.20, 0.25, 0.30]$ for each task based on the performance of the baseline safe RL algorithm CRPO compared to the unconstrained algorithm MPO. Figure 3 shows total rewards throughout training for each task across this range of safety coefficients. We selected the most difficult cost definition in this range (i.e., lowest safety coefficient value) where CRPO is still able to achieve the same total rewards as MPO (or the value that leads to the smallest gap between the two in the case of Walker Run and Quadruped Run). The resulting safety coefficients used for our experiments are listed in Table 2.

In order to evaluate the robustness of our learned policies, we generate a range of test environments for each task based on perturbing a simulator parameter in the Real-World RL Suite. See Table 3 for the perturbation parameters and corresponding ranges considered in our experiments. The test range for each domain is centered around the nominal parameter value that defines the single training environment used for all experiments except domain randomization. See Figure 4 for detailed results of the risk-averse and risk-neutral versions of our RAMU framework across all tasks and environment perturbations.

**Domain randomization**  Domain randomization requires a training distribution over a range of environments, which is typically defined by considering a range of simulator parameters. For the in-distribution version of domain randomization considered in our experiments, we apply a uniform distribution over a subset of the test environments defined in Table 3. In particular, we consider the middle 50% of test environment parameter values centered around the nominal environment value for training. In the out-of-distribution version of domain randomization, on the other hand, we consider a different perturbation parameter from the one varied at test time. We apply a uniform distribution over a range of values for this alternate parameter centered around the value in the nominal environment. Therefore, the only environment shared between the set of test environments and the set of training environments used for out-of-distribution domain randomization is the nominal environment. See Table 4 for details on the parameters and corresponding ranges used for training in domain randomization.

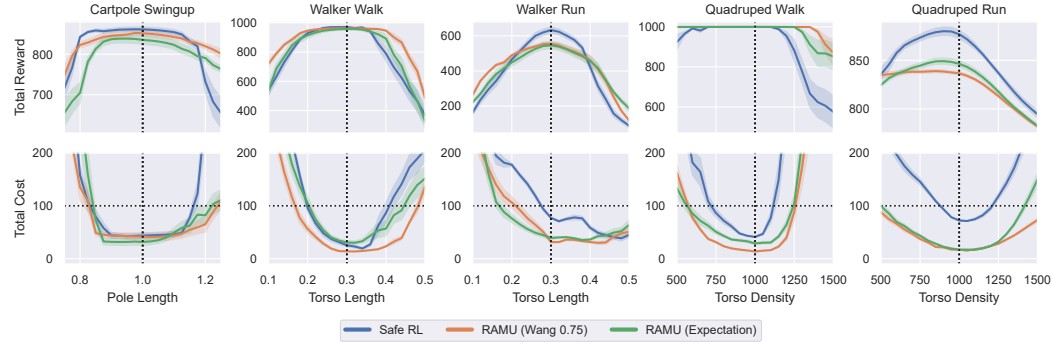

Figure 4: Comparison with standard safe RL across tasks and test environments. RAMU algorithms use the risk measure in parentheses applied to both the objective and constraint. Shading denotes half of one standard error across policies. Vertical dotted lines represent nominal training environment. Top: Total reward. Bottom: Total cost, where horizontal dotted lines represent safety budget.

Table 4: Perturbation parameters and ranges for domain randomization across domains

| Domain | Perturbation Parameter | Nominal Value | Training Range |
|---|---|---|---|
| In-Distribution | | | |
| Cartpole | Pole Length | 1.00 | $[0.875, 1.125]$ |
| Walker | Torso Length | 0.30 | $[0.20, 0.40]$ |
| Quadruped | Torso Density | 1,000 | $[750, 1,250]$ |
| Out-of-Distribution | | | |
| Cartpole | Pole Mass | 0.10 | $[0.05, 0.15]$ |
| Walker | Contact Friction | 0.70 | $[0.40, 1.00]$ |
| Quadruped | Contact Friction | 1.50 | $[1.00, 2.00]$ |

We include the results for domain randomization across all tasks and environment perturbations in Figure 5. Across all tasks, we observe that our RAMU framework leads to similar or improved constraint satisfaction compared to in-distribution domain randomization, while only using one training environment. In addition, our framework consistently outperforms out-of-distribution domain randomization, which provides little benefit compared to standard safe RL due to its misspecified training distribution.

**Adversarial reinforcement learning** In order to implement the action-robust PR-MDP framework, we must train an adversarial policy. We represent the adversarial policy using the same structure and neural network architecture as our main policy, and we train the adversarial policy to maximize total costs using MPO. Using the default setting in Tessler et al. [46], we apply one adversary update for every 10 policy updates.

We include the results for adversarial RL across all tasks and environment perturbations in Figure 6, where adversarial RL is evaluated without adversarial interventions. We see that adversarial RL leads to robust safety in some cases, such as the two Quadruped tasks. However, in other tasks such as Cartpole Swingup, safety constraint satisfaction is not as robust. Safety also comes at the cost of conservative performance in some tasks, as evidenced by the total rewards achieved by adversarial RL in Walker Run and Quadruped Run. Overall, our RAMU framework achieves similar performance to adversarial RL, without the drawbacks associated with adversarial methods that preclude their use in some real-world settings.

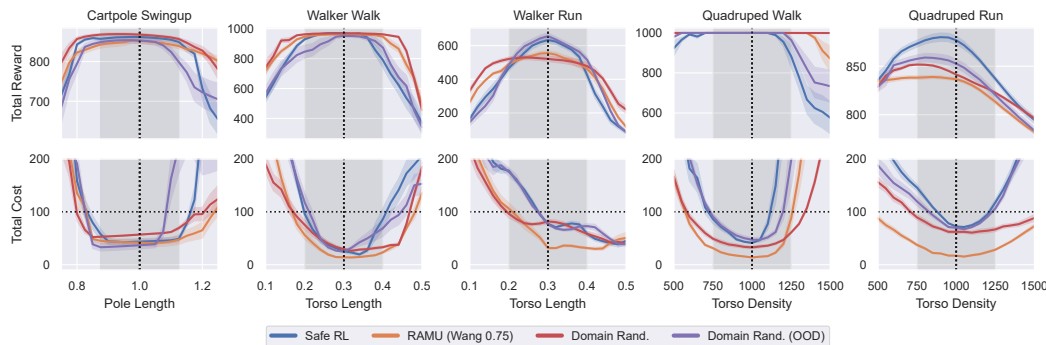

Figure 5: Comparison with domain randomization across tasks and test environments. Grey shaded area denotes the training range for in-distribution domain randomization. Shading denotes half of one standard error across policies. Vertical dotted lines represent nominal training environment. Top: Total reward. Bottom: Total cost, where horizontal dotted lines represent safety budget.

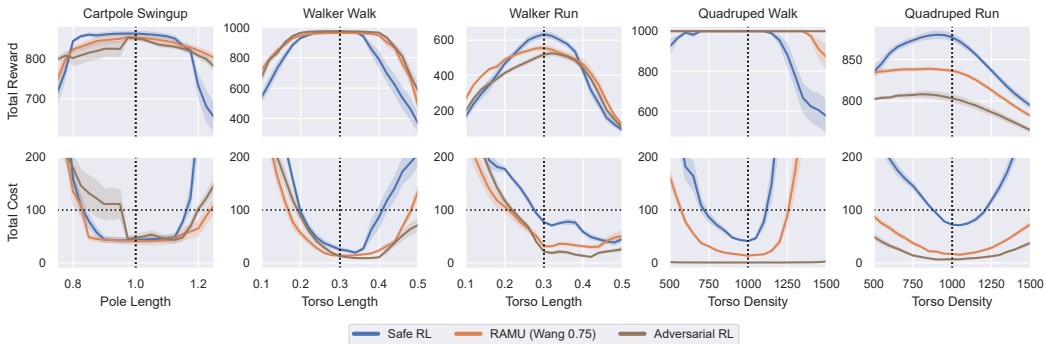

Figure 6: Comparison with adversarial RL across tasks and test environments. Performance of adversarial RL is evaluated without adversarial interventions. Shading denotes half of one standard error across policies. Vertical dotted lines represent nominal training environment. Top: Total reward. Bottom: Total cost, where horizontal dotted lines represent safety budget.

**Network architectures and algorithm hyperparameters** In our experiments, we consider neural network representations of the policy and critics. Each of these neural networks contains 3 hidden layers of 256 units with ELU activations. In addition, we apply layer normalization followed by a tanh activation after the first layer of these networks as proposed in Abdolmaleki et al. [2]. We consider a multivariate Gaussian policy, where at a given state we have $\pi(a \mid s) = \mathcal{N}(\mu(s), \Sigma(s))$ where $\mu(s)$ and $\Sigma(s)$ represent outputs of the policy network. $\Sigma(s)$ is a diagonal covariance matrix, whose diagonal elements are calculated by applying the softplus operator to the outputs of the neural network. We parameterize the reward and cost critics with separate neural networks. In addition, we consider target networks that are updated as an exponential moving average with parameter $\tau = 5\text{e-}3$.

We consider CRPO [55] as the baseline safe RL algorithm in all of our experiments, which immediately switches between maximizing rewards and minimizing costs at every update based on the value of the safety constraint. If the sample-average estimate of the safety constraint for the current batch of data satisfies the safety budget, we update the policy to maximize rewards. Otherwise, we update the policy to minimize costs.

After CRPO determines the appropriate objective for the current batch of data, we apply MPO [1] to calculate policy updates. MPO calculates a non-parametric policy update based on the KL divergence parameter $\epsilon_{\text{KL}}$, and then takes a step towards this non-parametric policy while constraining the KL divergence from updating the mean by $\beta_\mu$ and the KL divergence from updating the covariance matrix

Table 5: Network architectures and algorithm hyperparameters used in experiments

| General | |
|---|---|
| Batch size per update | 256 |
| Updates per environment step | 1 |
| Discount rate ($\gamma$) | 0.99 |
| Target network exponential moving average ($\tau$) | 5e-3 |

| Policy | |
|---|---|
| Layer sizes | 256, 256, 256 |
| Layer activations | ELU |
| Layer norm + tanh on first layer | Yes |
| Initial standard deviation | 0.3 |
| Learning rate | 1e-4 |
| Non-parametric KL ($\epsilon_{\mathrm{KL}}$) | 0.10 |
| Action penalty KL | 1e-3 |
| Action samples per update | 20 |
| Parametric mean KL ($\beta_\mu$) | 0.01 |
| Parametric covariance KL ($\beta_\Sigma$) | 1e-5 |
| Parametric KL dual learning rate | 0.01 |

| Critics | |
|---|---|
| Layer sizes | 256, 256, 256 |
| Layer activations | ELU |
| Layer norm + tanh on first layer | Yes |
| Learning rate | 1e-4 |

| RAMU | |
|---|---|
| Transition model samples per data point ($n$) | 5 |
| Latent variable hyperparameter ($\epsilon$) | 0.10 |

by $\beta_\Sigma$. We consider per-dimension KL divergence constraints by dividing these parameter values by the number of action dimensions, and we penalize actions outside of the feasible action limits using the multi-objective MPO framework [2] as suggested in Hoffman et al. [20]. In order to avoid potential issues related to the immediate switching between reward and cost objectives throughout training, we completely solve for the temperature parameter of the non-parametric target policy in MPO at every update as done in Liu et al. [28]. See Table 5 for the default hyperparameter values used in our experiments, which are based on default values considered in Hoffman et al. [20].

For our RAMU framework, the latent variable hyperparameter $\epsilon$ controls the definition of the distribution $\mu_{s,a}$ over transition models. Figure 7 shows the performance of our RAMU framework in Walker Run and Quadruped Run for $\epsilon \in [0.05, 0.10, 0.15, 0.20]$. A larger value of $\epsilon$ leads to a distribution over a wider range of transition models, which results in a more robust approach when combined with a risk-averse perspective on model uncertainty. We see in Figure 7 that our algorithm more robustly satisfies safety constraints as $\epsilon$ increases, but this robustness also leads to a decrease in total rewards. We consider $\epsilon = 0.10$ in our experiments, as it achieves strong constraint satisfaction without a meaningful decrease in rewards. Finally, for computational efficiency we consider $n = 5$ samples of transition models per data point to calculate sample-based Bellman targets in our RAMU framework, as we did not observe meaningful improvements in performance from considering a larger number of samples.

**Computational resources**  All experiments were run on a Linux cluster with 2.9 GHz Intel Gold processors and NVIDIA A40 and A100 GPUs. The Real-World RL Suite is available under the Apache License 2.0. We trained policies for 1 million steps across 5 random seeds, which required

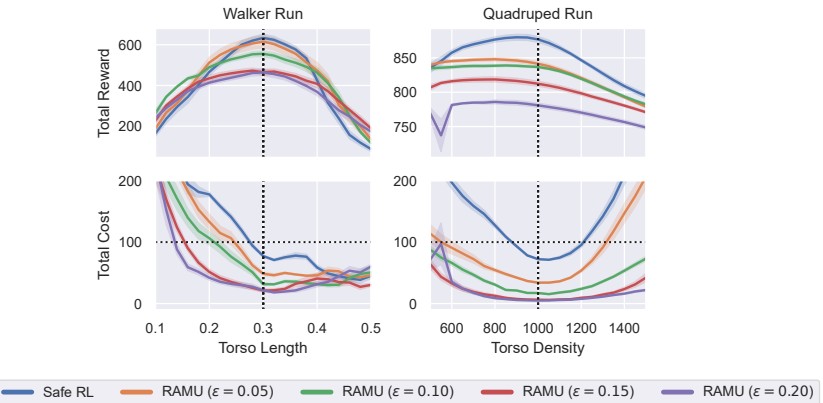

Figure 7: Hyperparameter sweep of latent variable hyperparameter $\epsilon$ on Walker Run and Quadruped Run. RAMU algorithms use the Wang transform with $\eta = 0.75$ applied to both the objective and constraint. Shading denotes half of one standard error across policies. Vertical dotted lines represent nominal training environment. Top: Total reward. Bottom: Total cost, where horizontal dotted lines represent safety budget.

approximately one day of wall-clock time on a single GPU for each combination of algorithm and task using code that has not been optimized for execution speed.

