# OpenReview forum: "Risk-Averse Model Uncertainty for Distributionally Robust Safe Reinforcement Learning"
_NeurIPS.cc/2023/Conference — NeurIPS 2023 poster_

### Official Review · Reviewer_JpRL · 2023-07-03

**Soundness:** 3 good
**Presentation:** 3 good
**Contribution:** 2 fair
**Rating:** 5
**Confidence:** 4

**Summary:**

The authors consider a distribution over transition models and tackle the safe RL problem by applying a risk-averse perspective towards model uncertainty through coherent distortion risk measures. The proposed formulation can ease the burden of solving a min-max problem, which is often encountered in many worst-case safe RL algorithms.

The authors also theoretically show that their formulation is equivalent to a specific class of distributionally robust safe RL problems.

**Strengths:**

- This work proposes a new formulation of safe RL, by considering a distribution of transition models and applying the distortion risk measure toward the model uncertainty, which circumvents the burden of solving min-max problems.
- This work theoretically proposes that the reformulated problem is equivalent to a specific class of distributionally robust safe RL problems.

**Weaknesses:**

- It is better to include the proof of Lemma 2 in Appendix B.2 for self-contained. Also, explaining the results of Lines 225-226 is better.
- It seems that the performance of the proposed method can not surpass existing methods, especially adversarial RL. Also, the experiments presented are not sufficient.
- Although applying the distortion risk measure $\rho$ seems to be promising, I'm wondering about the necessity of doing so since it introduces more complex computation processes and seems to contribute little to the experiment results.

**Questions:**

- What's the choice of the function $g$ (line 224-225) in your experiments?
- In lines 229-230, is there only one Q function for calculating all sampled transitions?
- In Table 1, what’s the meaning of the bold numbers? I think you should highlight the best results instead of yours.
- Also, please explain the necessity and merits of adopting the formulation proposed in this paper.

---

> ### Author Rebuttal · Authors · 2023-08-08
>
> Thank you for taking the time to review our paper. Please see below for detailed responses to your questions. In particular, we highlight the main benefits of our RAMU framework and the key takeaways of our experimental results. We hope that these responses address your main concerns, and we ask that you please consider updating your review scores to reflect these clarifications.
>
> ### [W3, Q4] Main benefits of our RAMU framework
>
> There are several benefits of our proposed RAMU framework compared to existing methods for addressing model uncertainty in deep RL (such as adversarial RL and domain randomization). The main benefits of our RAMU framework include:
>
> 1. Our RAMU framework has robustness guarantees (see Theorem 1). This is not true of popular methods such as domain randomization that apply the expectation operator over distributions of transition models.
> 2. Our RAMU framework achieves this robustness without requiring complex minimax optimization. This is not true of robust RL (and distributionally robust RL) methods, which must solve for worst-case transition models (or distributions over transition models) throughout training. Our approach, on the other hand, only requires weighted sample averages that can be computed very efficiently (see lines 220-235).
> 3. Our RAMU framework can be implemented using standard data collection from a single training environment (see lines 236-263), without requiring potentially dangerous adversarial interventions (as in adversarial RL) or detailed simulator access (as in domain randomization). Therefore, unlike these existing methods, our approach can be applied in settings that require real-world data collection for training.
>
> ### [W2, Q3] Key takeaways from experimental results
>
> * **[W2] Key takeaways:** We believe that our experimental results provide strong support for the benefits of our RAMU framework, and we include thorough comparisons against the most popular methods for addressing model uncertainty in deep RL. The experiments demonstrate the following key points:
>
>     1. Our RAMU framework achieves significant robustness and safety benefits compared to standard safe RL, while using exactly the same data collection process from a single training environment.
>     2. Our use of coherent distortion risk measures leads to robustness and safety benefits compared to a risk-neutral approach based on expectations, as shown by the comparison of RAMU (Wang 0.75) to the special case of RAMU (Expectation). Our risk-averse implementation achieves higher average rewards (1.08 vs. 1.05) and better safety constraint satisfaction (80% vs. 74%).
>     3. Our RAMU framework achieves similar or improved performance compared to the most popular baselines for incorporating model uncertainty in deep RL (domain randomization and adversarial RL). We accomplish this without requiring additional assumptions on the training process, such as detailed simulator access (as in domain randomization) or potentially dangerous adversarial interventions (as in adversarial RL), that are not always suitable in real-world settings.
>
> * **[W2] Comparison to adversarial RL:** Our RAMU framework achieves similar results to adversarial RL, and has significant benefits compared to adversarial RL in terms of implementation. RAMU achieves higher average rewards at test time in 3 out of 5 tasks (and higher average rewards in aggregate: 1.08 vs. 1.05), and better safety constraint satisfaction at test time in 3 out of 5 tasks (and similar safety constraint satisfaction in aggregate: 80% vs. 82%). Unlike adversarial RL, RAMU accomplishes this in a way that (i) does not alter the data collection process, (ii) does not require training an adversary in a minimax formulation, and (iii) does not require different implementations during training and testing. These all represent meaningful drawbacks of adversarial RL, which make adversarial RL unsuitable for training in many real-world tasks.
> * **[Q3] Table 1:** We bold the risk-averse implementation of our RAMU framework to highlight the relevant version of our approach for the reader to focus on.
>
> ### [Q1, Q2] Implementation details
>
> > [Q1] What's the choice of the function $g$ (line 224-225) in your experiments?
>
> In our experiments, we consider the Wang transform with $\eta = 0.75$ as well as the special (risk-neutral) case of the expectation operator (see lines 302-306). The function $g$ corresponding to these are shown in Figure 7 in Appendix C. The form of $g$ for the Wang transform is also in line 103, and the expectation operator corresponds to a linear $g$.
>
> > [Q2] In lines 229-230, is there only one Q function for calculating all sampled transitions?
>
> Yes, there is one RAMU cost Q function (and one RAMU reward Q function). As shown in the critic loss functions (lines 215-216), we are interested in calculating the estimates in (7) with the current target Q function. Therefore, for each sampled transition model, we apply the standard cost Bellman target in lines 229-230 using this single Q function. Then, we combine these estimates using the weighted sample average in (7).

---

> ### Comment · Area_Chair_f3R8 · 2023-08-16
> **Are you satisfied by the answers?**
>
> Dear reviewer,
>
> Would you please indicate whether the authors' response is satisfactory for you? If not, please engage with the authors, so we can get a better assessment of this work.
>
> Thank you,
> Area Chair

---

> > ### Comment · Area_Chair_f3R8 · 2023-08-18
> >
> > I am following up on this, especially given that your review is currently the most critical one. Do you find the authors response convincing or do you have a serious issue against this paper?
> >
> > Thank you,
> > Area chair

---

> > > ### Comment · Reviewer_JpRL · 2023-08-20
> > >
> > > I would like to thank the authors for their time to provide detailed responses to my concerns, and thus I'll increase my score to 5.
> > >
> > > The main advantage of adopting the proposed RAMU framework as claimed by the authors is its robustness guarantees and avoiding solving the min-max problem. However, I'm concerned by its actual contribution to solving practical tasks, since the empirical improvement is not significant to me as shown in Table 1. Also, I think you should bold the best method instead of your method in the experiment section.

---

> > > > ### Author Response · Authors · 2023-08-20
> > > >
> > > > Thank you for your response! In addition to the advantages that you have mentioned (main benefits #1 and #2 in our rebuttal), we also want to emphasize that our framework can be applied across a broad range of practical tasks because it requires minimal assumptions on the training process (see main benefit #3 in our rebuttal). Our approach only requires standard data collection from a single training environment, so it can even be applied in settings that require real-world data collection for training. This is not true of the popular baselines that we compare against in our experiments. Domain randomization requires detailed simulator access during training, and adversarial RL requires potentially dangerous adversarial interventions throughout training. These assumptions can be undesirable or impractical in many real-world settings. Our RAMU framework achieves similar or improved performance compared to these popular baselines, without requiring these additional assumptions on the training process (see item #3 under the key takeaways bullet in our rebuttal).

---

### Official Review · Reviewer_22QQ · 2023-07-06

**Soundness:** 3 good
**Presentation:** 3 good
**Contribution:** 2 fair
**Rating:** 6
**Confidence:** 5

**Summary:**

This paper proposes a methodology for distributionally robust RL via the use of risk measures and leveraging risk (Fenchel) duality, dealing with what they call model uncertainty. The paper introduces the RAMU Q function and Bellman operators respectively, which are based on modifying the standard risk-neutral operators by inducing risk-awareness over the choice of the transition model. The RAMU Q function and respective Bellman operators exhibit nice properties, due to the class of risk measures considered (distortion risk measures).

Then, the paper describes a model-free implementation of the proposed approach with a single training environment, leveraging interesting results for statistical estimation of risk functionals (specifically distortion risk measures). Lastly, the proposed approach is verified on a number of numerical benchmarks.

**Strengths:**

Overall, I think this is a paper with potential, and it was a pleasure reading it. The RAMU framework is indeed interesting and seems to be effective, although I have some concerns (see below).

The estimation of the Bellman operators in Section 6 within the RAMU framework can indeed be computed efficiently, as the authors point out, which is an advantage of the approach. The application of the results of Jones and Zitikis is quite interesting.

The experimental section is quite detailed and the comparisons with existing methods is well-thought, especially with a large number of similar alternative approaches. Also the experiments bring out the effect and efficacy of risk-averse RL well.



**Weaknesses:**

Risk duality and equivalence to minimax (distributionally robust) optimization (under certain assumptions) is very well known and established (in essence it is just Fenchel duality). Therefore, I do not think it can be claimed as novelty or contribution of this paper by itself (e.g., in lines 5 to 7 in the abstract). I would suggest that the authors rephrase to highlight more specifically their contributions.

In the discussion about risk measures, I do not see any explanation on *why* one should choose to work with distortion risk measures or coherent risk measures, and why the axioms proposed by Majumdar and Pavone are appropriate. Axioms are subjective, so in principle there is no universal reason to use one set of axioms over another set of axioms. Of course distortion risk measures exhibit amazing analytical properties and probably this is among the main reasons justifying the authors chosing this class, but the discussion and point needs to be made.

The RAMU cost function as defined in line 156 is introduced somewhat arbitrarily, in the sense that it is unclear if this is somewhat related to a base problem such as (2). While I understand the motivation and indeed it *seems* to make sense, usually one would start from such a base problem, possibly risk-averse in a certain way that makes intuitive and operational sense, and then build their way towards Q factors and Bellman optimality conditions.

Theorem 1 follows directly from the application of risk duality. I am not sure if this result "deserves" to be a theorem. I would think that a stating as a proposition would be more suiting. As a general comment, the theory in this paper is somewhat limited, however I have enjoyed reading the paper still.

The discussion under "Generative distribution of transition models", line 236 onwards is quite convoluted and involved without particular reason in my opinion. While I finally understood what happens (I think!), I believe that the authors should make an effort to write this part in a much clearer manner.



**Questions:**

What is a "distribution over transition models" in lines 41-42? This is very ambiguous at this point.

Also, related in Lines 72-74: Defining this product is somewhat obscure. For instance, does this make sense when S and A are uncountable sets (or you implicitly assume finite states and actions throughout)?

**Limitations:**

Yes

---

> ### Author Rebuttal · Authors · 2023-08-08
>
> Thank you for taking the time to review our paper. We are glad that you enjoyed reading it, and found our RAMU framework to be interesting and effective. We also appreciate your thoughtful suggestions, which will help to improve our paper. Please see below for responses to all of your questions and comments, which we hope address your main concerns. If so, we ask that you please consider updating your review scores to reflect our responses.
>
> ### [W1, W4] Theoretical contributions / novelty
>
> * Our RAMU framework represents a novel contribution to the RL literature, and we provide important theoretical results for the corresponding RAMU Bellman operators and Q functions that support the use of this framework. We provide theoretical connections to distributionally robust RL (Theorem 1), and we prove contraction properties for the RAMU Bellman operators (Corollary 2) that provide theoretical support for training the RAMU Q functions via standard temporal difference methods. It was not trivial to construct a novel RL framework for addressing model uncertainty with these theoretical properties and an efficient implementation.
> * We agree that the general connection between coherent risk measures and robustness is known (we include this known result in Appendix B.2). However, we apply this known result to establish an equivalence between our RAMU framework and a class of distributionally robust safe RL problems, which is a novel and important result. By establishing this connection, we are able to efficiently address model uncertainty in a deep RL context through the use of risk measures. We will update the paper to make this contribution clear, and we will include the distributionally robust safe RL problem definition to which our RAMU problem is equivalent.
>
> ### [W2] Choice of coherent distortion risk measures
>
> * In our RAMU framework, we leverage properties of **coherent** risk measures to provide robustness guarantees, and we leverage properties of **distortion** risk measures to provide an efficient, model-free implementation based on weighted sample averages that does not involve minimax optimization. We will add this commentary to the paper to make our choice of coherent distortion risk measures more clear.
> * Please see Majumdar and Pavone (2020) for a detailed discussion on why each of the properties of coherent distortion risk measures is important in the context of robotics. In general, coherent risk measures and distortion risk measures are two of the most popular classes of risk measures used in the literature, so there is a consensus that the characteristics of these classes are desirable.
>
> ### [W3, W5, Q1, Q2] Other clarifications
>
> > [W3] The RAMU cost function as defined in line 156 is introduced somewhat arbitrarily…usually one would start from a base problem…
>
> We will update Section 4 to start from the formal problem definition that corresponds to the RAMU update in (4). The objective and constraint in this formulation are very similar to the corresponding Q functions defined in line 156, just averaged over initial states and actions.
>
> > [W5] The discussion under "Generative distribution of transition models", line 236 onwards is quite convoluted and involved…
>
> The main goal of this section is to describe how we can define a distribution $\mu$ over transition models in a way that can be implemented using only data collected from a single training environment. We accomplish this by constructing samples from perturbed versions of the training environment. We will update this section to make this goal clear.
>
> > [Q1] What is a “distribution over transition models” in lines 41-42? This is very ambiguous at this point.
>
> “Distribution over transition models” in lines 41-42 refers to $\mu$, the same concept of a distribution over potential environments that was introduced in the previous paragraph (lines 34-35). We will update the language to make this more clear at this stage of the paper.
>
> > [Q2] …in Lines 72-74: Defining this product is somewhat obscure…
>
> This structure in lines 72-74 is known as rectangularity. It is a very common assumption throughout the robust RL literature (see line 75 for references), and we use standard notation from the literature in lines 72-74 to describe this structure. Rectangularity simply implies that the model uncertainty $\mu_{s,a}$ at every state-action pair is independent from other state-action pairs. In general, this is a conservative assumption, but it allows for recursive definitions of Q functions.

---

> ### Comment · Area_Chair_f3R8 · 2023-08-16
> **Are you satisfied by the answers?**
>
> Dear reviewer,
>
> Would you please indicate whether the authors' response is satisfactory for you? If not, please engage with the authors, so we can get a better assessment of this work.
>
> Thank you,
> Area Chair

---

> > ### Comment · Reviewer_22QQ · 2023-08-16
> >
> > I would like to thank the authors for taking the the time and providing detailed responses to my comments. I am mostly satisfied with the answers provided, except possibly with the point regarding using coherent or risk measures in [W2]. While the first bullet is solid, since such classes of risk measures have exceptional properties, the second bullet is highly subjective (for instance, there is a huge debate in the finance literature about the axioms comprising the class of coherent risk measures, and there is absolutely no consensus. It happens that such nice classes of risk measures are used in academic research, because of analytical tractability, which again is a very solid reason why to use them).
> >
> > For now, I will keep my score unchanged, but I am leaning favorably towards this paper.

---

### Official Review · Reviewer_mRCw · 2023-07-08

**Soundness:** 2 fair
**Presentation:** 3 good
**Contribution:** 3 good
**Rating:** 6
**Confidence:** 4

**Summary:**

This paper presents a Temporal Difference (TD) learning method for addressing the ``Risk-Averse Model Uncertainty for Distributionally Robust Safe Reinforcement Learning'' problem. Specifically, the authors consider a Constrained Markov Decision Process (CMDP) combined with Bayesian uncertainty sets. They are trying to attain a policy which is both robust and safe; Meaning that it optimizes a nested risk measure of discounted return, while satisfying certain guarantees over discounted cost. The proposed method is implemented and evaluated by comparing its performance against adversarial RL, Safe RL, and domain randomization approaches across multiple environments.

**Strengths:**

- This paper addresses a noteworthy problem. As a general matter, I believe that risk-aware policy selection methods for Bayesian MDP, as also studied in [1, 2, 3] is an interesting and open area of research.
- Even though the problem is notation-heavy, the paper is well presented and maintains a high level of readability.
- The authors have proven their implementation's effectiveness across five different tasks. Furthermore, the authors' utilization of a large number of training samples adds credibility to the algorithm's performance evaluation.


[1] Giorgio Angelotti, Nicolas Drougard, & Caroline Ponzoni Carvalho Chanel. (2023). An Offline Risk-aware Policy Selection Method for Bayesian Markov Decision Processes. arXiv preprint arXiv:2105.13431.

[2] Lobo, E. A., Ghavamzadeh, M., & Petrik, M. (2020). Soft-robust algorithms for batch reinforcement learning. arXiv preprint arXiv:2011.14495.

[3] Petrik, M., & Russel, R. H. (2019). Beyond confidence regions: Tight Bayesian ambiguity sets for robust MDPs. Advances in neural information processing systems, 32.

**Weaknesses:**

- **Theoretical results**: The primary focus of this paper is the implementation and experimental evaluation; there are limited theoretical results in this paper.

- **Comparisons**: In the ``strengths'' section of my review, I mentioned three risk-aware policy selection methods for Bayesian MDP methods that I am aware of [1, 2, 3]. These methods have similar objectives, albeit having two key differences (i) Instead of CMDPs the objective is for normal MDP and does not require safety guarantees (ii) the risk measure is (might be) applied over entire stochasticy $\mu$ rather than recursively over each $\mu_{s, a}$. As both algorithms use samples of models to have a Monto Carlo estimation of the risk measure, the $\theta$ update in Algorithm 1 is particularly similar to the $\operatorname{RiskEvaluation}$ operation introduced by [1].

    - Firstly, I would suggest that the authors include these papers in the ``uncertainty in reinforcement learning'' part of the related works.

    -  Secondly, it would have been more informative if the paper compared its proposed method to one of these existing methods, instead of focusing on comparisons with adversarial RL and domain randomization.

- **Presentation**:  Theorem 1 requires a more concrete and explicit formulation. $\zeta_{a, s}$ is introduced as ``depending on $\rho^+$ and $\rho$''; without properly defining the dual presentation of coherent risk measures beforehand. This lack of clarity makes it very challenging for non-expert readers to understand the paper.


In conclusion, while the paper proposes an interesting problem and provides valuable experimental results, there is a lot of room for improvement. Thus, I rate this paper as a borderline accept.

**Questions:**

- **The Objective**: It would be interesting to observe the performance of the proposed algorithm by varying the learning rates for $\theta_r$ and $\theta_c$ and exploring different risk measures for $\rho^+$ and $\rho$. For instance imagine we set $\rho^+$ as $CVaR_{\alpha_r}^+$, and $\rho$ as $CVaR_{\alpha_c}$. As $\alpha_c$ decreases, the algorithm becomes more strict on violating safety conditions, and as $\alpha_r$ decreases, the algorithm becomes more robust. I am curious to see the differences in experimental outcomes under such settings.
- **Evaluation**: In the experiments, different algorithms are being compared based on the safety percentage, and total rewards. I would like to see their comparison based on (an estimated version of) the Lagrangian form of the actual objective (4) (while fixing the set of hyperparameters for all methods). Since the proposed algorithm directly aims to optimize this objective, it is reasonable to expect it to outperform other methods in terms of this specific evaluation metric. The conduct of such an evaluation can serve as a sanity check.

**Limitations:**

no limitations

---

> ### Author Rebuttal · Authors · 2023-08-08
>
> Thank you for taking the time to review our paper. We are glad you agree this is an important problem, and we appreciate your comments on our paper’s clear presentation and strong experimental results. Please see below for responses to your questions. We hope that these clarifications address your main concerns. If so, we ask that you please consider updating your review scores to reflect our responses.
>
> ### [W1, W3] Theoretical results / presentation
>
> * **[W1] Theoretical results:** Our RAMU framework represents a novel contribution to the RL literature, and we provide important theoretical results for the corresponding RAMU Bellman operators and Q functions that support the use of this framework. We provide theoretical connections to distributionally robust RL (Theorem 1), and we prove contraction properties for the RAMU Bellman operators (Corollary 2) that provide theoretical support for training the RAMU Q functions via standard temporal difference methods. It was not trivial to construct a novel RL framework for addressing model uncertainty with these theoretical properties and an efficient implementation.
> * **[W3] Presentation:** Due to space constraints, we present Theorem 1 with the necessary details to provide an intuitive understanding of the result, and we defer the full formal treatment of the result to the Appendix. Appendix B.2 formally defines the appropriate probability space and provides the dual representation result that we use (including reference to the appropriate dual space). We will update Theorem 1 in the main text for additional clarity.
>
> ### [W2] Comparison of RAMU to other methods
>
> * **Experiment baselines:** In our experiments, we compare against the most popular methods for robustness to model uncertainty in deep RL. Adversarial RL represents a common implementation of robust RL in the deep RL setting, so we use a popular action-robust adversarial RL method as a baseline in our experiments. Domain randomization is the most popular implementation based on distributions of transition models, so we also consider this as a baseline.
> * **Comparison to [1, 2, 3]:** We agree that [1, 2, 3] are interesting works that consider model uncertainty in RL. However, they are not well suited as baselines in the deep RL setting that is the focus of this paper. The methods in [1, 2, 3] have several key differences compared to the setting that we consider in this paper. Most importantly, they are offline RL methods (vs. online RL in this paper) that are designed for tabular settings (vs. deep RL in this paper). Please also note that we do not focus on a Bayesian setting in this work (however, by choosing $\mu$ to represent a posterior, our framework can be applied in a Bayesian setting).
>
> ### [Q1, Q2] Experimental analysis
>
> * **[Q1]:** The comparison between RAMU (Wang 0.75) and RAMU (Expectation) represents an example of the analysis you suggest, as the expectation operator is equivalent to setting the Wang hyperparameter to zero. An alternative way to change the level of robustness is to vary the hyperparameter $\epsilon$ that defines our distribution $\mu$. We include these results in Appendix C (Figure 6), which demonstrates the trends you have mentioned. We expect that varying a risk measure hyperparameter would lead to similar trends (see comparison between RAMU with Wang 0.75 vs. Expectation).
> * **[Q2]:** The goal of safe RL is to maximize rewards while satisfying the safety constraint. Therefore, we directly measure these two key quantities as our metrics of interest in our experiments. We have also included the total rewards and total costs for every algorithm and test environment in Appendix C (Figures 3, 4, 5). Note that a Lagrangian relaxation would be one way to solve the safe RL problem, but we do not consider this approach in our experiments (we use CRPO; see lines 654-658 for details).

---

> > ### Comment · Reviewer_mRCw · 2023-08-16
> > **Response**
> >
> > The authors have responded to most of my empirical concerns. Therefore, I will increase my rating from 5 to 6.
> >
> > Regarding Q1: First it was only a suggestion, but comparison between RAMU (Wang 0.75) and RAMU (Expectation) does not cover what I mentioned there. It is only for 2 risk measures, and also does not investigate different $\alpha_c$ and $\alpha_r$.

---

> ### Comment · Area_Chair_f3R8 · 2023-08-16
> **Are you satisfied by the answers?**
>
> Dear reviewer,
>
> Would you please indicate whether the authors' response is satisfactory for you? If not, please engage with the authors, so we can get a better assessment of this work.
>
> Thank you,
> Area Chair

---

### Official Review · Reviewer_ZX7b · 2023-07-09

**Soundness:** 3 good
**Presentation:** 3 good
**Contribution:** 3 good
**Rating:** 6
**Confidence:** 2

**Summary:**

The paper introduces a deep reinforcement learning framework for safe decision-making in uncertain environments. The authors propose a risk-averse approach towards model uncertainty using coherent distortion risk measures. They provide robustness guarantees for the framework by showing its equivalence to a distributionally robust safe reinforcement learning problem. The framework is efficient and model-free, utilizing standard data collection from a single training environment. Experiments on continuous control tasks with safety constraints demonstrate the framework's robust and safe performance across a range of perturbed test environments.

**Strengths:**

[+] The paper is logically clear and well-written, making it easy to follow.

[+] The paper starts from theory, first defining a new Q-function and a new Bellman operator, then proving the equivalence between the new Bellman operator and distributionally robust Bellman operators with respective ambiguity sets, thereby verifying the rationality of the definition and the contraction property of the new Bellman operator. This provides a theoretical basis for the algorithm design.

[?] What is the relationship between safety and robustness? The paper theoretically proves that the newly proposed RAMU Bellman operator has robustness guarantees. However, it seems not to mention much about its relationship with Constrained Markov Decision Process (CMDP) and how to ensure that the point converged to by the RAMU Bellman operator is safe.

[?] Related to the previous question, why consider both safety and robustness here? It seems that considering both safety and robustness is crucial for the Theorem 1. Is it necessary to clarify this point more explicitly in the abstract? For example, the current abstract's first sentence emphasizes the importance of safety, the line 5 separately emphasizes robustness guarantees, and the line 7 re-emphasizes robustness, which can be somewhat confusing.



**Weaknesses:**

see above

**Questions:**

see above

**Limitations:**

see above

---

> ### Author Rebuttal · Authors · 2023-08-08
>
> Thank you for taking the time to review our paper. We are glad you found the paper to be clear and well-written, and appreciated the theoretical support for our proposed framework. Please see below for responses to your questions, which clarify the importance of both safety and robustness. If these clarifications address your concerns, we ask that you please consider updating your overall review score to reflect this.
>
> ### [Q1, Q2] Importance of safety and robustness
>
>  * Safety is often a prerequisite for real-world decision making applications, so we consider a safe RL setting with a Constrained MDP (CMDP) as our starting point. However, safe RL with a CMDP finds a policy that is only safe in a single training environment, with no robustness guarantees related to performance and safety in other environments. In many real-world scenarios, the environment at deployment time may be different from the training environment due to factors such as modeling errors or unknown disturbances.
> * We incorporate robustness to model uncertainty in both the objective (rewards) and safety constraint (costs) of a CMDP. We accomplish this by learning separate RAMU Q functions for the reward and cost, which both appear in our RAMU update in (4). As demonstrated in our experimental results, this update leads to policies that (i) achieve robust performance (due to the use of our RAMU reward Q function in the objective) and (ii) remain safe across a range of test environments (due to the use of our RAMU cost Q function in the safety constraint). Our RAMU framework significantly outperforms the standard safe RL baseline that applies the update in (2), which uses standard Q functions that only consider a single training environment $p$ and do not incorporate robustness.

---

> ### Comment · Area_Chair_f3R8 · 2023-08-16
> **Are you satisfied by the answers?**
>
> Dear reviewer,
>
> Would you please indicate whether the authors' response is satisfactory for you? If not, please engage with the authors, so we can get a better assessment of this work.
>
> Thank you,
> Area Chair

---

> > ### Comment · Area_Chair_f3R8 · 2023-08-18
> >
> > Following up on this!

---

### Official Review · Reviewer_F3rP · 2023-07-19

**Soundness:** 2 fair
**Presentation:** 3 good
**Contribution:** 2 fair
**Rating:** 5
**Confidence:** 4

**Summary:**

The paper introduces the Risk-Averse Model Uncertainty (RAMU) framework for safe reinforcement learning in uncertain environments. RAMU incorporates a distribution of transition models and applies a risk-averse perspective using coherent distortion risk measures. The framework offers an efficient, model-free implementation through one single training environment. Experimental results demonstrate the framework's ability to produce robust, safe performance in perturbed test environments. Unlike existing distributional robust (DR) approaches, RAMU eliminates the need for minimax optimization.

**Strengths:**

- The proposed perturbation function models the distribution of environment transition models, providing a foundation for practical implementation of distributionally robust algorithms.
-  Compared to existing methods, the RAMU framework is implemented efficiently. It avoids complex minimax optimization, which is the major obstacle to applying Distributional Robustness methods to DRL.


**Weaknesses:**

- The problem addressed in this paper is the handling of model uncertainty in safe reinforcement learning (RL) scenarios. However, it appears that the issue of model uncertainty in safe RL is similar to the problem in standard RL. Since the proposed RAMU method is not specifically designed for safe RL, it would be more convincing if it were compared to existing distributionally robust methods such as those discussed in [1] and [2].
- The formulation of the distributionally robust safety problem in the article lacks clarity. The problem definition starts with directly modifying the Q function to the DR Q function in Eq4, which is a shortcut approach. It would have been more appropriate to first define the problem and then derive the suitable form of the Q function. Although solving the Eq4 definition is straightforward by plugging in the safety RL method with DR RL, it is important to note that the worst-case transitions associated with the reward and cost (the $\beta$ in Eq5 and Eq6) are not the same. However, in reality, there is only one transition, which makes the proposed method more conservative in practice.
-  The transition function f(s, s') mentioned in line 257 may only be effective for tasks involving robot control, where dynamics follow linear patterns and such perturbations are effective. For tasks involving image inputs, it remains unclear what kind of function f should be used.

**Questions:**

- I understand that the contribution of the paper lies in proposing a distributionally robust (DR) safe reinforcement learning method that does not require a minmax operation. However, given that there is no direct connection between DR algorithm design and safety, it is essential to compare it with existing DR methods such as [1] and [2].
- The proposed algorithms does have similarities to policy smoothing algorithms, as mentioned in [3] and [4]. It is plausible to consider that combining policy smoothing with adversarial RL could potentially yield better results compared to the RAMU framework. In other words, exploring the combination of observational robustness and action robustness may provide improvements over predefined model robustness?

Ref.

[1] Robust Reinforcement Learning using Offline Data

[2] Distributionally Robust Q-Learning

[3] Deep Reinforcement Learning with Robust and Smooth Policy

[4] Policy Smoothing for Provably Robust reinforcement learning.

**Limitations:**

As mentioned in the Conclusion section of the paper, the choice of the model distribution µ and risk measure ρ in the RAMU framework is user-defined, and RAMU framework only addresses robustness with respect to model uncertainty and safety defined by expected total cost constraints. Crruently, this paper only consider the dynamics of model transition follow linear patterns, which may not to be suitable for tasks that involve image inputs.

---

> ### Author Rebuttal · Authors · 2023-08-08
>
> Thank you for taking the time to review our paper. Please see below for clarifications and responses to your questions. In particular, we clarify the definition of “distributionally robust RL” [a, b] considered in our theoretical results, and how this differs from other definitions that have appeared in the literature more recently. We hope that our responses address your main concerns, and we ask that you please consider updating your review scores to reflect these clarifications.
>
> ### [W1, Q1, Q2] Distributionally robust RL and comparison to other methods
>
> * **Definition of distributionally robust RL:** We consider a distribution $\mu$ over transition models in our work, and show in Theorem 1 that our RAMU framework is equivalent to a class of distributionally robust RL problems as defined by [a, b]. Distributionally robust RL [a, b] considers ambiguity sets of *distributions over transition models* in $P(\mathcal{M})$ (see lines 79-81), and is different from robust RL [c, d] that applies uncertainty sets directly over transition models in $\mathcal{M}$.
> * **[W1, Q1] Comparison to [1] and [2]:** Please note that both [1] and [2] consider the typical robust RL setting [c, d] based on uncertainty sets of transition models. This is different from our approach, which considers a distribution over transition models and is equivalent to a class of distributionally robust RL problems as defined by [a, b]. In recent years, researchers have started using “robust” and “distributionally robust” interchangeably to mean robust RL [c, d], which has caused some confusion. Also different from our approach, [1] considers the offline RL setting and [2] focuses on the tabular RL case.
> * **[Q2] Comparison to [3] and [4]:** We agree that the idea of observational robustness considered in [3] and [4] is an important area of research, but it is not the focus of this work. We focus on being robust to uncertainty in the transition model (i.e., dynamics), and these two sources of uncertainty require different analysis and algorithms. Please note that [3] and [4] do not make any direct connections to the definition of distributionally robust RL [a, b] considered in this work. Action robustness is more closely related to uncertainty in dynamics because it leads to changes in state transitions, which has led to its use as a robust RL [c, d] method in deep RL settings.
> * **Experiment baselines:** We compare against the most popular methods for robustness to model uncertainty in deep RL. Adversarial RL represents a common implementation of robust RL [c, d] in the deep RL setting, so we use a popular action-robust adversarial RL method as a baseline in our experiments. Domain randomization is the most popular implementation based on distributions of transition models, so we also consider this as a baseline.
>
> ### [W2] Problem formulation
>
> * We will update Section 4 to start from the formal problem definition that corresponds to the RAMU update in (4). The objective and constraint in this formulation are very similar to the corresponding Q functions defined in line 156, just averaged over initial states and actions. We will also make clear the distributionally robust safe RL problem definition to which our RAMU problem is equivalent, which involves ambiguity sets of distributions over transition models (see lines 79-81).
> * As you pointed out, the worst-case distributions $\beta$ over transition models will be different for rewards and costs in our formulation. Because we do not know the true environment at test time, it makes sense to take this conservative approach in order to guarantee robustness in both the rewards and costs at deployment time. This is a common approach when considering robustness in a safe RL setting [e].
>
> ### [W3] Extension to image inputs
>
> As an example that makes sense for our experiments, we consider an intuitive perturbation function based on percentage changes in each dimension of state transitions. However, our methodology works with any choice of distribution $\mu$ over transition models (or equivalently, any choice of perturbation function $f_x$), which is defined by the user to best suit the application. In RL from images, it is common to consider an MDP in a latent representation space, and we can apply our methodology in this latent space. Alternatively, in scenarios where detailed simulator access is available, it would also be possible to generate next state images from multiple transition models by leveraging this simulator.
>
> **References:**
>
> [a] H. Xu and S. Mannor. Distributionally robust Markov decision processes. In Advances in Neural Information Processing Systems, volume 23. Curran Associates, Inc., 2010.
>
> [b] P. Yu and H. Xu. Distributionally robust counterpart in Markov decision processes. IEEE Transactions on Automatic Control, 61(9):2538–2543, 2016.
>
> [c] G. N. Iyengar. Robust dynamic programming. Mathematics of Operations Research, 30(2): 257–280, 2005.
>
> [d] A. Nilim and L. E. Ghaoui. Robust control of Markov decision processes with uncertain transition matrices. Operations Research, 53(5):780–798, 2005.
>
> [e] D. J. Mankowitz, D. A. Calian, R. Jeong, C. Paduraru, N. Heess, S. Dathathri, M. Riedmiller, and T. Mann. Robust constrained reinforcement learning for continuous control with model misspecification. arXiv preprint, 2021. arXiv:2010.10644.

---

> > ### Comment · Reviewer_F3rP · 2023-08-14
> >
> > I believe the core contribution of this article lies in providing a simple and effective robust method to address model uncertainty in MDPs. However, the method itself does not directly address the safety aspect or the CMDP problem. The article needs to clarify the connection between safety and robustness. While the article only addresses the robustness issue, the overall background of the article is strongly tied to the concept of safety, which seems strange to me. Additionally, there is significant room for improvement in the writing of this article. However, I would like to increase the score to 5 fot the good idea of distribution over transition models.

---

> > > ### Author Response · Authors · 2023-08-14
> > >
> > > Thank you for your response! We agree that the core contribution of our paper is a simple and effective method for incorporating robustness to model uncertainty in a deep RL setting.
> > >
> > > We consider safe RL modeled by a Constrained MDP (CMDP) as our starting point, as safety is often a prerequisite for real-world decision making applications. We incorporate robustness to model uncertainty in the safe RL setting by applying our RAMU framework to both the objective (rewards) and safety constraint (costs) of a CMDP. As demonstrated in our experimental results, this leads to policies that achieve robust performance *and* robust safety across test environments (i.e., robustness in both components of a CMDP). If safety is not relevant in an application, our framework could also be applied to provide robustness in a standard MDP, which would result in a special case of the update in (4) without the safety constraint.
> > >
> > > In the updated version of our paper, we will clarify how our problem formulation leads to robust performance *and* robust safety constraint satisfaction in a CMDP. Thank you for helping us to improve our paper.

---

### Author Rebuttal · Authors · 2023-08-08

Thank you to all of the reviewers for their thoughtful feedback. We are excited to see the reviewers agree that the paper is clear and well-written (ZX7b, mRCw, 22QQ), proposes a novel framework with a practical and efficient implementation (F3rP, 22QQ, JpRL), and provides strong theoretical (ZX7b, JpRL) and experimental (mRCw, 22QQ) support for this framework. We have replied directly to each reviewer with detailed responses, and we will update the paper to incorporate clarifications based on reviewer suggestions. If we have addressed your main concerns, we ask that you please consider updating your review scores to reflect our responses. Thank you for helping us to improve our paper!

---

### Decision · Program_Chairs · 2023-09-21

**Decision:**

Accept (poster)

**Comment:**

The paper introduces the RAMU framework for safe reinforcement learning in uncertain environments. It uses a distribution of transition models and a risk-averse approach with coherent distortion risk measures. The authors aim to create a policy that's both robust and safe by combining Constrained Markov Decision Processes (CMDP) with distribution over uncertainty sets, optimizing a nested risk measure of discounted return while meeting cost-related guarantees. RAMU is efficient and model-free, working within a single training environment.

All reviewers are positive about this paper. We have two Borderline Accepts and three Weak Accepts. Among the positive aspects of this paper are
- Novelty of the framework for safe RL.
- Theoretical results
- Efficient implementation that avoids complex minimax optimization.

There are some concerns too, including (not comprehensive list):
- Reviewers request clarification on the connection between safety and robustness in the article.
- Missing some relevant work.
- Complicated discussion at some points.

Overall, I believe this is a good paper. **Given the unanimous support of all reviewers, I recommend its acceptance.**